

# Bogoliubov phonons in a Bose-Einstein condensate from the one-loop perturbative renormalization group

**Niklas Rasch**[1⋆]**, Aleksandr N. Mikheev**[1,2] **and Thomas Gasenzer**[1,2]

**1** Kirchhoff-Institut für Physik, Ruprecht-Karls-Universität Heidelberg,
Im Neuenheimer Feld 227, 69120 Heidelberg, Germany
**2** Institut für Theoretische Physik, Ruprecht-Karls-Universität Heidelberg,
Philosophenweg 16, 69120 Heidelberg, Germany

⋆ niklas.rasch@kip.uni-heidelberg.de

## Abstract

Wilson's renormalization-group approach to the weakly-interacting single-component Bose gas is discussed within the symmetry-broken, condensate phase. Extending upon the work by Bijlsma and Stoof [1], wave-function renormalization of the temporal derivative contributions to the effective action is included in order to capture sound-like quasiparticle excitations with wave lengths larger than the healing-length scale. By means of a suitable rescaling scheme we achieve convergence of the coupling flows, which serve as a means to determine the condensate depletion in accordance with Bogoliubov theory, as well as the interaction-induced shift of the critical temperature.

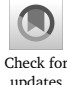

# 1 Introduction

Since the experimental realization of Bose-Einstein condensation in dilute interacting atomic gases [2,3] there has been great interest in further exploring these systems both experimentally and theoretically. The basic theoretical approach is based on the Gross-Pitaevskii model, which, at the relevant low energies, describes the weak interparticle interactions within a contact-potential approximation. To capture the condensate phase and its fundamental excitations, Bogoliubov's ansatz builds on the assumption of a spontaneously broken U(1) symmetry and on an expansion about the field expectation value [4]. A subsequent truncation at Gaussian order represents an approximation, in which Bogoliubov-de Gennes diagonalization of the propagator yields the spectrum of quasiparticle excitations, which have collective, sound-wave character at long wavelengths. It allows determining leading-order perturbative corrections to thermodynamic observables such as the depletion of the condensate density and the associated shift of the chemical potential [5]. In three spatial dimensions, the approximation is valid in the dilute-gas regime, where the characteristic $s$-wave scattering length is much smaller than the interparticle spacing. Beyond this order of approximation, perturbation theory allows computing further corrections. At next-to-leading order, Beliaev's approximation involves all one-loop corrections to the Bogoliubov quasiparticle effective action [6–8]. See [9] for a review of related approaches to the weakly interacting Bose gas.

In the aforementioned perturbative extensions of the self-consistent framework, the sound-wave energy of excitations, linear in momentum $k$ for $k \to 0$, typically gives rise to logarithmic infrared (IR) divergences in $d = 3$, which makes it difficult to apply self-consistent diagrammatic renormalization methods [10]. It was shown that these divergences cause

the anomalous self-energy to vanish at zero momentum [11, 12], which implies that self-consistent higher-order approaches require the corresponding normal contribution to the zero-momentum self-energy to equal the chemical potential in order for the Hugenholtz-Pines or Goldstone theorem to be fulfilled and thus the dispersion to be gapless [13]. Different approaches have been explored to obtain, in this context, perturbative corrections, including self-consistent diagrammatic perturbation theory [11,12,14,15], in density-phase representation [16], within a number-conserving perturbative approach [17,18], or in a both, conserving and gapless Hartree-Fock-Bogoliubov approximation [19]. In the strong-coupling limit a resummation scheme of IR divergences has been presented in [20]. IR divergences can in general be handled by means of renormalization group (RG) theory [21–26]. Within the RG approach, one iteratively integrates out fluctuations starting from the ultraviolet (UV) end of the spectrum and reabsorbs their effect in redefined coupling constants. This is achieved by means of RG flow equations governing the couplings, which parametrize an effective description at a given maximum spatial resolution scale.

The weakly interacting Bose gas in $d = 3$ and thermal equilibrium has been extensively studied using RG methods [1, 27–52]: Near criticality, the interaction-induced shift in the critical temperature has been determined using flow equations in perturbative approximation [1,36,39], within a non-perturbative functional RG approach [44,45,50], and in the high temperature limit [29,37,38,40,41]. The results can be compared with those of numerical approaches [53–58]. Furthermore, RG treatments have served to obtain critical exponents such as the exponent $\nu$ governing the scaling of the correlation length near criticality. Thereby, different techniques and approximation methods were employed, including derivative expansions [30], Wilson's momentum-shell method [1], and expansions in powers of $\epsilon = 4 - d$ in $d$ spatial dimensions [34], considering homogeneous as well as confined [31,32,36] systems. In [1], the critical exponent was determined as $\nu = 0.685$. Given that the thermal condensation phase transition of the interacting Bose gas belongs to the O(2) universality class, this value is to be compared with high-precision results from both theoretical and experimental studies. An expansion of anomalous dimensions up to six loops in perturbation theory in three spatial dimensions gave $\nu = 0.6703(15)$, while the $\epsilon$-expansion, which avoids the problem of IR divergences, yields $\nu = 0.6680(35)$ [59]. Measuring the superfluid phase transition in liquid $^4$He gave a value of $\nu = 0.6717(4)$ [60], cf. [61] for a review.

The RG treatment presented here is not focused on computing universal properties near criticality. It is rather motivated by the divergent flows of the chemical potential $\mu$ and of the interaction coupling $g$ described in [1], which were obtained from a Taylor expansion with respect to the slowly varying modes of the one-loop effective action in leading-order derivative expansion, i.e., in local-potential approximation. In fact, these divergences can be traced to a violation of the U(1) symmetry at a given order of approximation, and imposing the relevant Ward identities to be satisfied can be avoided [28,35]. At order of approximation chosen in [1], the RG flow does not account for the system entering the linear, phononic regime of the Bogoliubov dispersion, such that finite coupling flows are achieved only within the short-wave-length regime $\Lambda_{\text{th}}^2 \ll (na_0)^{-1}$, where the linear part of the dispersion relation is negligible. Here, $n$ is the particle density, $a_0$ the $s$-wave scattering length and $\Lambda_{\text{th}}^2 = 2\pi/(mT)$ the thermal de Broglie wavelength.

So far, the description of linear, phononic excitations has been achieved within a functional RG approach both at $T = 0$ [42–44, 46, 47, 49] and $T \neq 0$ [44, 45, 48, 51, 52], within a next-to-leading order gradient expansion $\mathcal{O}(\nabla^2, \partial_\tau, \partial_\tau^2)$ of the effective action. The vanishing anomalous self energy has been linked to the transition to an effective SO($d + 1$) space-time symmetry in the IR [42, 43]. Approaching the low temperature limit from the thermal phase transition [44,45,48] the thermodynamic bulk quantities approach the Bogoliubov expressions in the limit of low temperatures [45, 52]. Beyond the above approaches, using an interpola-

tion between the cartesian and amplitude-phase representations of the Bose field, converging flows and the hydrodynamic effective action valid in the IR were determined in [51, 52].

Extending upon [1] and using the framework developed by Wilson [21–24], we here derive perturbative, one-loop RG flow equations at next-to-leading order in the derivative expansion, that take into account renormalization of the first- and second-order temporal derivative terms in the action functional, neglecting, for the first, the renormalization of the kinetic energy term. This allows us to obtain a description of the transition of a dilute Bose gas in $d = 3$ dimensions to the IR phononic regime with a linear sound-wave dispersion. In this broken-symmetry regime, we make use of the Hugenholtz-Pines relation, which results from a Ward identity reflecting the U(1) symmetry of the model. In this respect, our approach is similar to that of [28,35] for a $T = 0$ system. As we remain within the perturbative, one-loop approximation and neglect wave-function renormalization of the spatial derivative terms, the scope of our analysis can not be to improve upon precision beyond results from numerical analyses or functional RG approaches, but to present a simple perturbative RG approach to sound excitations in a Bose condensate. Besides giving results consistent with the Bogoliubov-de Gennes theory, this approach can be considered as a first step towards treating more complex problems such as non-universal as well as universal properties in a multi-component dilute Bose gas [62,63].

Our paper is organized as follows. In Sect. 2, we introduce the Wilsonian effective action of the Gross-Pitaevskii model in next-to-leading order gradient expansion, for which we derive, in Sect. 3, the RG flow equations both, in the symmetric, 'thermal', and the symmetry-broken, condensate phases. Our numerical results are presented in Sect. 4, including a comparison of the flows with and without wave-function renormalization, and a determination of the depletion of the condensate and the interaction-induced shift of the critical temperature. We summarize and draw our conclusions in Sect. 5.

## 2 Weakly interacting Bose gas

### 2.1 Gross-Pitaevskii model

A single-component gas of weakly interacting bosons of mass $m$, exposed to an external (e.g. trapping) potential $V_{ext}$, is described by a complex field $\Psi(\mathbf{x})$, the classical dynamics of which is governed by the Gross-Pitaevskii (GP) Hamiltonian

$$H = \int d^d x \left( -\Psi(\mathbf{x})^* \frac{\nabla^2}{2m} \Psi(\mathbf{x}) + V_{ext}(\mathbf{x})|\Psi(\mathbf{x})|^2 + \frac{g}{2}|\Psi(\mathbf{x})|^4 \right). \tag{1}$$

Throughout this paper we use natural units where $\hbar = k_B = 1$. The quartic coupling term accounts for two-body scattering, which can be described, at low energies, as a contact interaction. In three spatial dimensions, $d = 3$, the coupling $g$ is related to the $s$-wave scattering length $a_0$ by $g = 4\pi a_0/m$. Genuine three-body and higher interactions can be neglected in the dilute limit, sufficiently far from a Feshbach resonance, as their effect on the later renormalization is negligible [1,64]. Since in ultracold gases, the total particle number in a closed volume $V$,

$$N = \int_V d^d x \, |\Psi(\mathbf{x})|^2 = \int d^d x \, n(\mathbf{x}), \tag{2}$$

is conserved, one usually resorts to a grand canonical ensemble, in which the chemical potential $\mu$ acts as a Lagrange multiplier in the 'Kamiltonian' $K = H - \mu N$. Within the Bogoliubov mean-field approach, the classical field configuration in the ground-state of (1) is then determined, in the absence of an external trapping potential, by the stationary GP equation for the

field expectation value $\psi = \langle \Psi \rangle$:

$$\left( -\frac{\nabla^2}{2m} - \mu + g\,|\psi(\mathbf{x})|^2 \right)\psi(\mathbf{x}) = 0\,. \tag{3}$$

Assuming positive $g$, i.e., repulsive interactions, the mean-field ground state, at non-zero density $n = |\psi|^2$, is uniform, $n(\mathbf{x}) \equiv n_0 = n_c = \text{const}$, and thus the chemical potential $\mu$, in the condensed phase, becomes a positive energy, resulting, in this mean-field approximation, as $\mu = g n_c$. The density enters the non-vanishing field expectation value $\psi = \sqrt{n_c}\,\mathrm{e}^{\mathrm{i}\phi}$, with a constant but irrelevant phase angle $\phi$, which constitutes spontaneous breaking of the U(1) symmetry of the model Hamiltonian (1).

## 2.2 Effective action in next-to-leading order gradient expansion

In view of the renormalization group (RG) approach to the quantized system in thermal equilibrium, one introduces the (Wilsonian) effective action $S$,

$$S[\Psi] = \int_x \left( \Psi^*\big(Z_\tau \partial_\tau - V \partial_\tau^2 - Z_x \frac{\nabla^2}{2m} - \mu\big)\Psi + \frac{g}{2}|\Psi|^4 \right), \tag{4}$$

which follows from the coherent-state path integral of (1). The fluctuating fields also depend on the imaginary time $\tau$, which is integrated, within $\int_x \equiv \int_0^\beta \mathrm{d}\tau \int \mathrm{d}^d x$, from $0$ to $\tau = \beta = 1/T$, with periodic boundary conditions imposed along the time direction on the bosonic fields, $\Psi(0, \mathbf{x}) = \Psi(\beta, \mathbf{x})$. As a consequence, the action can be expanded in terms of discrete Matsubara frequency modes $\omega_n = 2\pi n T$, cf. (A.1) in App. A. The Gaussian part of (4) yields

$$S_0[\Psi_k] = \sum_k \!\!\!\!\!\!\!\!\!\!\!\!\!\!\!\! \Psi_k^*\big(-\mathrm{i}Z_\tau \omega_n + V\omega_n^2 + Z_x \varepsilon_k - \mu\big)\Psi_k\,, \tag{5}$$

where the shorthands $\sum_k = \beta^{-1}\sum_{\omega_n} \int \mathrm{d}^d k\,/(2\pi)^d$ and $\Psi(\omega_n, \mathbf{k}) = \Psi(k) \equiv \Psi_k$ are used and the single-particle energy $\varepsilon_k = \mathbf{k}^2/(2m)$ is introduced. For the interaction part of (4) one finds

$$S_{\mathrm{int}}[\Psi_k] = \frac{g}{2}\sum_{k_i} \!\!\!\!\!\!\!\!\!\!\!\!\!\!\!\! \delta(k_2 + k_4 - k_1 - k_3)\,\Psi_{k_1}^* \Psi_{k_2} \Psi_{k_3}^* \Psi_{k_4}\,, \tag{6}$$

where the $\delta$-distribution $\delta(k) = (2\pi)^d \beta\, \delta_{0,\omega_n} \delta(\mathbf{k})$ ensures energy-momentum conservation.

In analogy to existing schemes, the Berry-phase term $\sim \Psi^* \partial_\tau \Psi$ as well as the kinetic energy receive couplings $Z_\tau$ and $Z_x$, respectively, commonly termed wave-function renormalization factors. Within the gradient expansion, we furthermore introduce a second-order time derivative term $\sim \Psi^* \partial_\tau^2 \Psi$ with the coupling $V$. The latter explicitly breaks the Galilean symmetry of (4) at $T = 0$ which is already broken for $T > 0$ [43, 44]. The RG flow is initialized by the leading order action in the gradient expansion, where the couplings take their microscopic values in the UV, i.e., $Z_{\tau,\mathrm{in}} = 1$, $V_{\mathrm{in}} = 0$, and $Z_{x,\mathrm{in}} = 1$.

While renormalization gives rise to and alters all couplings that are allowed by symmetry, our choice is motivated by the aim to explicitly account for a linear sound-wave dispersion in the low-temperature condensate. In this regime, the first-order time derivative gets replaced by a second-order one, i.e., $Z_\tau \to 0$, while $V$ grows with the RG flow; and, neglecting residual sound-wave interactions, the system assumes an approximate Euclidean SO($d+1$) symmetry in space-time, as previously described in the functional RG framework [42–44]. The coupling $Z_x$, away from criticality in $d \geq 2$ dimensions, is anticipated to neither qualitatively nor quantitatively modify our results [44,52]. We therefore defer the determination of its flow to future work and keep it here for bookkeeping purposes as it gets rescaled during the flow, cf. Sect. 3.3.

## 2.3 Symmetry-broken phase

Before we move on to the WRG formulation, we introduce and discuss a parametrization of the effective action, which takes into account spontaneous symmetry breaking in the Bose-condensed phase. The quasiparticle excitations can be approximated in a Gaussian fashion by means of a saddle-point expansion of the action (4). In Bogoliubov theory, one expresses the field in terms of fluctuations around its expectation value, $\Psi(x) = \sqrt{n_c} + \delta\Psi(x)$, and only retains terms up to quadratic order and solves the resulting action exactly. Setting, for the first, $V = 0$ and $Z_x = Z_\tau = 1$, one finds an excitation frequency spectrum of gapless Bogoliubov modes,

$$\omega_k = \sqrt{\varepsilon_k(\varepsilon_k + 2gn_c)}, \tag{7}$$

that also follows directly from a linearization of the GP equation (3). Taking into account the possibility of $V \neq 0$, a second Bogoliubov mode emerges, which will be discussed in the next section.

Within the WRG formulation, one similarly needs to break the U(1) symmetry by hand, choosing an ansatz which assumes the presence of a condensate. The U(1) symmetry of the action (4) is broken by splitting the field $\Psi_k = \langle\Psi_k\rangle + \psi_k$ into its condensate part $\langle\Psi_k\rangle$ and thermal fluctuations $\psi_k \equiv \psi(k)$, with $\langle\psi_k\rangle = 0$. Performing a saddle-point expansion, the vanishing of the linear term implies, without external potential, the condensate field to be a uniform background $\langle\Psi_k\rangle = \delta(k)\psi_c = \delta(k)\sqrt{n_c}$, with $n_c$ being the condensate density, and the chemical potential to result as $\mu = n_c g$. This relation will remain valid under the renormalization procedure. As before, we dropped the irrelevant phase of the condensate.

The expansion furthermore gives rise to a modified Gaussian part of the action,

$$\widetilde{S}_0 = \sum_k \left[ \psi_k^*\left(-iZ_\tau\omega_n + V\omega_n^2 + Z_x\varepsilon_k + \mu_1\right)\psi_k + \frac{\mu_2}{2}\left(\psi_k\psi_{-k} + \psi_k^*\psi_{-k}^*\right) \right] + \sqrt{n_c}\left(n_c g - \mu\right)\left(\psi_0 + \psi_0^*\right), \tag{8}$$

where we have dropped constant terms as they correspond to irrelevant shifts of the energy zero.

Through the lens of renormalization, couplings are defined for each field term; thus, two generalized two-point couplings $\mu_1$ and $\mu_2$, with initial values

$$\mu_{1,\text{in}} = 2n_c g - \mu, \qquad \mu_{2,\text{in}} = n_c g, \tag{9}$$

appear in the quadratic part of the action. These couplings are equal initially, $\mu_{1,\text{in}} = \mu_{2,\text{in}}$, and remain so throughout the RG flow, thereby ensuring a gapless dispersion relation in line with Goldstone's and, equivalently, the Hugenholtz-Pines theorem [13]. This will need to be proven explicitly (cf. Sect. 3.2.3 below), before which we treat them as in general independent. The terms proportional to $\mu_2$ in (8) give rise to non-vanishing anomalous correlators $\langle\psi\psi\rangle_0$.

In the symmetry-broken phase, the interacting part of the action results as

$$\widetilde{S}_{\text{int}} = \sqrt{n_c}g\sum_{k_1,k_2,k_3}\delta(k_1 - k_2 - k_3)(\psi_1^*\psi_2\psi_3 + \text{c.c.}) + \frac{g}{2}\sum_{k_1,k_2,k_3,k_4}\delta(k_2 + k_4 - k_1 - k_3)\psi_1^*\psi_2\psi_3^*\psi_4. \tag{10}$$

Hence, three-point interactions emerge between one condensed and three thermal modes. This term is responsible for the appearance of wave-function renormalization at the one-loop level as discussed in the following.

## 3 Wilsonian RG flows

In this section, we summarize the WRG approach [21–24] as applied to Bose gases described by the action (4) and extend it to the symmetry-broken regime, cf. (8), (10). We derive the

flow equations in both the thermal and the condensed phase up to one-loop order. Wave-function renormalization in terms of $Z_\tau$ and $V$ is included straight from the beginning. Our notation follows Ref. [65].

The procedure of WRG starts with defining the action $S[\mathbf{g}]$ on a range of momenta $\lambda_0 \leq |\mathbf{k}| \leq \Lambda_0$ for the set of running couplings $\mathbf{g}$,

$$\mathbf{g} = \left(Z_x, Z_\tau, V, \mu, g\right). \tag{11}$$

The IR cut-off $\lambda_0$ refers, e.g., to the inverse linear size of the considered system. We can, in principle, take it to $\lambda_0 \to 0$, which we will do during the numerical evaluation of the RG equations.

The UV cut-off $\Lambda_0$ reflects the range of validity of the action on microscopic scales. To ensure the applicability of the $s$-wave interactions we choose it to be $\Lambda_0 \lesssim 1/a_0$. Note that, in the following, we are interested in the effective renormalization of the action to obtain a low-energy effective theory of the thermodynamic properties of a dilute ultracold Bose gas in the symmetry-broken, Bose-Einstein condensed phase. Hence, we will use WRG to study the flow of the effective couplings to momentum scales farther in the IR than $\Lambda_0$, rather than studying their renormalization in the farther UV. While the latter is used to express the couplings in terms of the microscopic scattering properties of, e.g., the alkali atoms, our aim here is to determine the effective couplings and field correlations as functions of thermodynamic bulk quantities such as particle density and temperature.

The Wilsonian renormalization flow is the result of many infinitesimal steps each consisting of three sub-steps:

(A) The first one is mode elimination, in which an infinitesimal momentum shell $\Lambda \leq |\mathbf{k}| \leq \Lambda_0$, with

$$b = \Lambda_0/\Lambda \to 1^+, \tag{12}$$

is integrated out from the partition function. The effective couplings $\mathbf{g}$, Eq. (11), defining the action on the remaining momentum volume $\lambda_0 \leq |\mathbf{k}| \leq \Lambda$ are thereby being modified.

(B) In the subsequent rescaling step, the renormalized couplings of the new effective action defined on the remaining momentum range are obtained by rescaling all couplings by powers of $b$. This is necessary to arrive at a new, coarse-grained effective description of the system valid at correspondingly larger length scales.

(C) In the final step, one combines both, steps (A) and (B) to update the couplings.

The three steps are then repeated in an iterative manner to determine a flow of the couplings with the continuously decreasing cut-off scale $\Lambda$.

## 3.1 Mode elimination: Symmetric phase

In the following, the above steps are described in more detail and specific to the Bose gas in the symmetric, 'thermal' phase. The mode elimination step starts with splitting the field $\Psi_k$ into a long-wavelength (IR) $\Psi_k^<$ and a short-wavelength (UV) $\Psi_k^>$ part:

$$\Psi_k = \Psi_k^< + \Psi_k^> = \Theta(\Lambda - |\mathbf{k}|)\Psi_k + \Theta(|\mathbf{k}| - \Lambda)\Psi_k. \tag{13}$$

They are separated by the cut-off $\Lambda$ above which the short wavelengths will be subsequently integrated out. Inserting this into the action $S[\Psi_k]$ one derives a part below the cut-off $\Lambda$, $S^<[\Psi_k^<]$, one above the cut-off, $S^>[\Psi_k^>]$, as well as a mixed term $S_{\text{mix}}^{<>}[\Psi_k^<, \Psi_k^>]$. Due to the lacking overlap, $\Psi_k^< \Psi_k^> = 0$, the Gaussian part of the action $S_0$ splits into a sum $S_0 = S_0^< + S_0^>$.

Only the interaction part contains mixing of the IR and UV scales,

$$S_{\text{int}} = S_{\text{int}}^< + S_{\text{int}}^> + g \sum_{k_i} \delta(k_2 + k_4 - k_1 - k_3) \tag{14}$$

$$\times \left( \Psi_1^{*<} \Psi_2^< \Psi_3^{*<} \Psi_4^> + \frac{1}{2} \Psi_1^{*<} \Psi_2^> \Psi_3^{*<} \Psi_4^> + \Psi_1^{*<} \Psi_2^< \Psi_3^{*>} \Psi_4^> + \Psi_1^{*<} \Psi_2^> \Psi_3^{*>} \Psi_4^> + \text{c.c.} \right),$$

where we use the shorthand $\Psi_{k_i} \equiv \Psi_i$. In the next step, the UV modes are integrated out explicitly from the partition function, which determines all thermodynamic bulk quantities:

$$\mathcal{Z} = \int \mathcal{D}\Psi^* \mathcal{D}\Psi \, e^{-S[\Psi; \mathbf{g}]} = \int \mathcal{D}\Psi^{*<} \mathcal{D}\Psi^< \, e^{-S^<[\Psi^<; \mathbf{g}^<]}. \tag{15}$$

We thus rewrite the partition function in terms of the macroscopic IR fields $\Psi^<$ only. The coarse-grained action $S^<[\Psi^<; \mathbf{g}^<]$ entering the partition function after mode elimination hence only depends on the modes $|\mathbf{k}| < \Lambda$. Since the effective action we aim to derive must exhibit the same thermodynamic behavior as the microscopic action, the partition function must not alter, and therefore the coupling constants $\mathbf{g}^<$ are suitably modified as compared to the original ones. While, so far, all the steps have in principle been exact, to determine the modified couplings one needs to solve the path integral in the UV over an action that contains higher-order than quadratic terms in $\Psi$ and $\Psi^*$. Since such integrals can in general not be computed analytically, we will make use of perturbation theory and write $S^<[\Psi^<; \mathbf{g}^<]$ as an expansion around $S_0^>[\Psi^>; \mathbf{g}]$ to second-order, which corresponds to a one-loop approximation:

$$S^<[\mathbf{g}^<] = S^<[\mathbf{g}] + \left\langle S_{\text{mix}}^{<>}[\mathbf{g}] + S_{\text{int}}^>[\mathbf{g}] \right\rangle_{0,>} + \frac{1}{2} \left[ \left\langle S_{\text{mix}}^{<>}[\mathbf{g}] + S_{\text{int}}^>[\mathbf{g}] \right\rangle_{0,>}^2 - \left\langle \left( S_{\text{mix}}^{<>}[\mathbf{g}] + S_{\text{int}}^>[\mathbf{g}] \right)^2 \right\rangle_{0,>} \right]. \tag{16}$$

Here, a constant term has been dropped, which corresponds to an irrelevant shift of the energy zero. $\langle \mathcal{O} \rangle_{0,>}$ denotes an expectation value with respect to the UV free action $S_0^>[\mathbf{g}]$, and we have dropped the explicit notion of the field arguments $\Psi^{\gtrless}$. To determine $\mathbf{g}^<$, the expectation values in (16) need to be expressed in powers of the IR field. The explicit change can be found after evaluating the free two-point correlators of $\Psi_k^>$. Note that the structure of the quadratic terms in (16) is such that only connected Feynman diagrams contribute to $\mathbf{g}^<$ [25]. This is reminiscent of the expansion of the free energy in terms of connected diagrams.

Free correlators are in general obtained by functional derivatives,

$$\left\langle \Psi_k^* \Psi_{k'} \right\rangle_0 = \frac{\delta}{\delta J_k} \frac{\delta}{\delta J_{k'}^*} \ln \mathcal{Z}_0[J] \bigg|_{J=0}, \tag{17}$$

of the Gaussian generating functional,

$$\mathcal{Z}_0[J] = \int \mathcal{D}\Psi^* \mathcal{D}\Psi \exp\left( -S_0[\mathbf{g}] + \sum_k \left[ J_k^* \Psi_k + \text{c.c.} \right] \right) = \mathcal{Z}_0[0] \exp\left( \sum_k \mathcal{J}_{-k}^T \mathcal{M}_k^{-1} \mathcal{J}_k \right), \tag{18}$$

with respect to the current $J_k$. The vectorial current $\mathcal{J}_k^T$ is defined in terms of the real and imaginary parts of the current expressed in spatial coordinates, i.e.,

$$\mathcal{J}_k^T = \left( [J_k + J_{-k}^*]/2, \quad [J_k - J_{-k}^*]/2i \right), \tag{19}$$

and the corresponding kernel matrix $\mathcal{M}_k$ is given by

$$\mathcal{M}_k = \begin{pmatrix} Z_x \varepsilon_k + V \omega_n^2 - \mu & -Z_\tau \omega_n \\ Z_\tau \omega_n & Z_x \varepsilon_k + V \omega_n^2 - \mu \end{pmatrix}. \tag{20}$$

The two-point normal and anomalous correlators are found using its inverse,

$$\mathcal{M}_k^{-1} = \frac{1}{\det(\mathcal{M}_k)} \begin{pmatrix} Z_x \varepsilon_k + V \omega_n^2 - \mu & Z_\tau \omega_n \\ -Z_\tau \omega_n & Z_x \varepsilon_k + V \omega_n^2 - \mu \end{pmatrix}, \tag{21}$$

and taking functional derivatives (17) of $\mathcal{Z}_0[J]$:

$$\langle \Psi_k^* \Psi_{k'} \rangle_0 = \frac{\delta(k-k')}{Z_x \varepsilon_k + V \omega_n^2 - \mu + iZ_\tau \omega_n} \equiv \delta(k-k') G(k), \tag{22}$$

$$\langle \Psi_k \Psi_{k'} \rangle_0 = 0. \tag{23}$$

Eq. (23) reflects the U(1) symmetry of the action (4), which ensures particle number conservation in the thermal phase. All higher-order expectation values in (16) can be decomposed by means of Wick's theorem into products of the two-point correlators (22).

In the following, we express the expansion (16) in Feynman diagrammatic form in terms of the free normal correlator $G(k)$, (22), and couplings $\mathbf{g}$. The expectation value $\langle S_{\text{int}}^>[\mathbf{g}] \rangle_{0,>}$ does not contain macroscopic fields $\Psi_k^<$ and thus consists of vacuum diagrams which only shift the energy zero. Furthermore, since all disconnected Feynman diagrams cancel in (16), we can discard $\langle S_{\text{mix}}^{<>}[\mathbf{g}] + S_{\text{int}}^>[\mathbf{g}] \rangle_{0,>}^2$.

As we neglect all two-loop diagrams, the remaining contributions to be evaluated are $\langle S_{\text{mix}}^{<>}[\mathbf{g}] \rangle_{0,>}$ and $\langle (S_{\text{mix}}^{<>}[\mathbf{g}])^2 \rangle_{0,>}$. Also diagrams contributing to the renormalization of the six-point coupling, i.e., three-particle scattering, are neglected since we are in the weakly-interacting, dilute regime.

The first of these terms gives the tadpole correction

$$\langle S_{\text{mix}}^{<>}[\mathbf{g}] \rangle_{0,>} = 2g \sum_{\substack{k,k' \\ \Lambda < |\mathbf{k}'| < \Lambda_0}} \Psi_k^{*<} G(k') \Psi_k^< \propto \sum_{\substack{k,k' \\ |\mathbf{k}| < \Lambda < |\mathbf{k}'| < \Lambda_0}} \quad . \tag{24}$$

Note that, in the first integral, we do not need to explicitly restrict the range of momenta $k$, as this is achieved by the $\Theta$-functions in $\Psi_k^<$, cf. (13). It is worth mentioning, that this diagram does not contain contributions to wave-function renormalization, since the correlator $G(k')$ is independent of the external momentum $k$. Hence, this term does only renormalize the coupling attributed to $\Psi_k^{*<} \Psi_k^<$, i.e., the chemical potential $\mu$. Generally speaking, at one-loop order in the symmetric phase no wave-function renormalization in $Z_\tau$ and $V$ emerges. Using the mode elimination equation (16), one can directly read off the new chemical potential entering $S^<[\mathbf{g}^<]$:

$$\mu^< = \mu - 2g \sum_{\substack{k \\ \Lambda < |\mathbf{k}| < \Lambda_0}} G(k). \tag{25}$$

The remaining expectation value in (16) contributes to the renormalization of the two-particle scattering:

$$\left\langle (S_{\text{mix}}^{<>}[\mathbf{g}])^2 \right\rangle_{0,>} \propto \quad . \tag{26}$$

Labeling the external momenta in (26) by $k_1, \ldots, k_4$ we obtain the integral

$$\left\langle (S_{\text{mix}}^{<>}[\mathbf{g}])^2 \right\rangle_{0,>} = \sum_{\substack{k_i, k \\ \Lambda < |\mathbf{k}| < \Lambda_0}} \delta(k_2 + k_4 - k_1 - k_3) \Psi_1^{*<} \Psi_2^< \Psi_3^{*<} \Psi_4^< g^2 G(k) \Big[ 4 G(k_4 - k_3 + k) + G(k_4 + k_2 - k) \Big]. \tag{27}$$

Note that, here, loop momentum depends on the external momenta, which generates momentum- and frequency-dependent interaction terms such as $\mathbf{k}^2\Psi^4$ and $\omega_n\Psi^4$. However, these couplings are neglected as they exhibit irrelevant scaling according to their engineering dimensions (cf. Sect. 3.3) and we thus only take the zeroth-order Taylor expansion around $k$ into account. Inserting (27) into (16) one reads off the new four-point coupling

$$g^< = g - \frac{g^2}{2} \oint_{\substack{k \\ \Lambda < |\mathbf{k}| < \Lambda_0}} \left[ 4\,G(k)\,G(k) + G(k)\,G(-k) \right].$$ (28)

The opposite-sign momenta in the last term correspond to the twofold scattering of two incoming particles with each other.

## 3.2 Mode elimination: Symmetry-broken phase

In this section, mode elimination is extended to the symmetry-broken, i.e., condensate phase. Mode splitting and integrating out the momentum shell between $\Lambda$ and $\Lambda_0$ proceeds as before, but now using the ansatz (8) and (10) for the effective action in evaluating $S^<[\mathbf{g}^<]$ as given in (16).

### 3.2.1 Normal and anomalous correlators

We begin again by determining the free correlator (17), as well as the anomalous one,

$$\langle \psi_k \psi_{k'} \rangle_0 = \frac{\delta}{\delta J_k^*} \frac{\delta}{\delta J_{k'}^*} \exp\left( \oint_k \mathcal{J}_{-k}^T \mathcal{M}_k^{-1} \mathcal{J}_k \right)\Bigg|_{\mathcal{J}=0}.$$ (29)

In the symmetry-broken phase, the kernel matrix $\mathcal{M}_k$ takes the from

$$\mathcal{M}_k = \begin{pmatrix} Z_x \varepsilon_k + V\omega_n^2 + \mu_1 + \mu_2 & -Z_\tau \omega_n \\ Z_\tau \omega_n & Z_x \varepsilon_k + V\omega_n^2 + \mu_1 - \mu_2 \end{pmatrix}.$$ (30)

The diagonals of the normal, $\langle \psi_k^* \psi_{k'} \rangle_0 = \delta(k - k') G_k$, and anomalous correlator, $\langle \psi_k \psi_{k'} \rangle_0 = \delta(k + k') \widetilde{G}_k$, result as

$$G_k = \frac{Z_x \varepsilon_k + \mu_1 + iZ_\tau \omega_n + V\omega_n^2}{V^2 \omega_n^4 + A_k \omega_n^2 + \omega_k^2}, \quad \text{and}$$

$$\widetilde{G}_k = -\frac{\mu_2}{V^2 \omega_n^4 + A_k \omega_n^2 + \omega_k^2},$$ (31)

respectively. Here, we introduced $A_k = Z_\tau^2 + 2V(Z_x \varepsilon_k + \mu_1)$ and the dispersion relation at $V = 0$:

$$\omega_k^2 = (Z_x \varepsilon_k + \mu_1)^2 - (\mu_2)^2 .$$ (32)

In the case of $\mu_1 = \mu_2 > 0$ this reduces to the gapless Bogoliubov dispersion (7). If $V \neq 0$, factorizing the denominator into $V^2[\omega_n^2 + (\omega_k^+)^2][\omega_n^2 + (\omega_k^-)^2]$ results in the quasiparticle frequencies as the poles of the correlators,

$$(\omega_k^\pm)^2 = \frac{A_k \pm \sqrt{A_k^2 - 4V^2 \omega_k^2}}{2V^2} .$$ (33)

In the limit $V \to 0$ one has $(\omega_k^-)^2 \to \omega_k^2/Z_\tau^2$ and $V^2(\omega_k^+)^2 \to Z_\tau^2$, and with the initial $Z_\tau = 1$ one recovers (32). The gapped dispersion relation $\omega_k^+$ is initially decoupled at $V = 0$.

The asymptotic behavior for $V \gg 1$ of $(\omega_k^+)^2 \to (Z_x \varepsilon_k + \mu_1 + \mu_2)/V$ and $(\omega_k^-)^2 \to (Z_x \varepsilon_k + \mu_1 - \mu_2)/V$ yields two relativistic dispersion relations. The same result is found for $Z_\tau \to 0$, which makes the action (4) quasi-relativistic and is equivalent to $V \gg 1$.

Hence, for $\mu_i \neq 0$, the dispersion relations describe the transition from the Bogoliubov dispersion at UV scales, $V \ll 1$, into the phononic regime, with a gapped and, for $\mu_1 = \mu_2$, a gapless mode in the $k \to 0$ limit.

### 3.2.2 Mode decomposition of the action

As in the symmetric phase, we need to compute the expectation values in (16). However, a bigger variety of Feynman diagrams appears owing to the anomalous correlator and the three-point vertex. Splitting the fields into $\psi = \psi^< + \psi^>$ leads, again, to $\widetilde{S}_0 = \widetilde{S}_0^< + \widetilde{S}_0^>$. The quartic interactions $\widetilde{S}_{\text{int}}^{(4)}$ decompose as in the thermal phase (14), while the cubic interactions decompose according to

$$\widetilde{S}_{\text{int}}^{(3)} = \sqrt{n_c} g \sum_{k_1, k_2, k_3} \delta(k_1 - k_2 - k_3) \left[ 2\psi_1^{*<} \psi_2^> \psi_3^< + 2\psi_1^{*>} \psi_2^> \psi_3^< + \psi_1^{*>} \psi_2^< \psi_3^< + \psi_1^{*<} \psi_2^> \psi_3^> + \text{c.c.} \right] + \widetilde{S}_{\text{int}}^{(3)<} + \widetilde{S}_{\text{int}}^{(3)>}. \tag{34}$$

Collecting all mixed contributions from (14) and (34) yields $\widetilde{S}_{\text{mix}}^{<>}$. Its expectation value reads

$$\left\langle \widetilde{S}_{\text{mix}}^{<>} \right\rangle_{0,>} = \frac{g}{2} \sum_{\substack{k,k' \\ \Lambda < |\mathbf{k}'| < \Lambda_0}} \left[ 4\psi_k^{*<} \psi_k^< G_{k'} + \left( \psi_k^< \psi_{-k}^< + \psi_k^{*<} \psi_{-k}^{*<} \right) \widetilde{G}_{k'} \right] + \sqrt{n_c} g \left( \psi_0^< + \psi_0^{*<} \right) \sum_{\substack{k \\ \Lambda < |\mathbf{k}| < \Lambda_0}} \left( 2G_k + \widetilde{G}_k \right). \tag{35}$$

In contrast to the situation in the thermal phase, the above expectation value does not suffice in determining the change in the two-point couplings because additional two-vertex diagrams from the second-order term must be taken into account. The above result does not feature wave-function renormalization since all loop correlators are independent of the external momentum, as it was the case in (24).

Note furthermore that the last term in (35) represents a tadpole contribution, which introduces a term linear in the fields to the action,

$$S_d[h] \propto h(\psi_0^< + \psi_0^{*<}). \tag{36}$$

The coupling constant $h$ vanishes in the broken-symmetry action (8) where $\mu = g n_c = g n$ is chosen initially. A renormalization of $h$ will account for a depletion of the condensate density due to zero-point and thermal excitations.

When calculating $\langle (\widetilde{S}_{\text{mix}}^{<>})^2 \rangle_{0,>}$, all disconnected diagrams can be dropped, since, as before, they mutually cancel in (16). We keep restricting our approximation to one-loop diagrams and up to quartic interactions, i.e., two-particle scattering. Following the approach employed in Ref. [1], we can restrict ourselves to determining the change in the two-point couplings, while the four-point coupling can be obtained from the constraint $\mu = n_c g$. This allows us to only keep terms that are quadratic in the IR fields, i.e., diagrams with two external legs:

$$\left\langle \left( \widetilde{S}_{\text{mix}}^{<>} \right)^2 \right\rangle_{0,>} = \left\langle \left( \widetilde{S}_{\text{mix}}^{(3)<>} \right)^2 \right\rangle_{0,>} \propto \text{—}\bullet\hspace{-2pt}\bigcirc\hspace{-2pt}\bullet\text{—} \,. \tag{37}$$

Here, energy-momentum conservation causes the loop momentum to depend on the external momentum. Hence, in the condensed phase, wave-function renormalization emerges already at one-loop order. One finds

$$\frac{1}{2}\left\langle\left(\widetilde{S}_{\text{mix}}^{(3)<>}\right)^2\right\rangle_{0,>} = n_{\text{c}}g^2\sum_{\substack{k,k'\\ \Lambda<|\mathbf{k}'|<\Lambda_0}}\left\{\psi_k^<\psi_k^{*<}\left[4\widetilde{G}_{k'}\widetilde{G}_{k'-k}+2G_{k'}\left(4\widetilde{G}_{k'-k}+2G_{k'-k}+G_{k-k'}\right)\right]\right.$$
$$\left.+\left(\psi_k^<\psi_{-k}^<+\psi_k^{*<}\psi_{-k}^{*<}\right)\left(2G_{k'-k}G_{k'}+3\widetilde{G}_{k'+k}\widetilde{G}_{k'}+4G_{k'}\widetilde{G}_{k+k'}\right)\right\}.\quad (38)$$

### 3.2.3 Coupling renormalization

We begin by considering the change in the local-in-momentum part of the quadratic part of the action. To that end, we Taylor expand the expressions (35) and (37) to the zeroth order in $k$. Terms beyond this order will be needed for the flow equations of the anomalous couplings $Z_\tau$ and $V$, which we will derive in 3.2.5 below.

Once both expectation values, (35) and (38), have been evaluated, we can read off $\delta h$ and $\delta\mu_i$. For this, one inserts these expectation values into (16) and, e.g., for $\mu_1$, collects all terms proportional to $\psi_k^<\psi_k^{*<}$, which then represent the modification $\delta\mu_1$. Doing so yields

$$\delta h = \sqrt{n_{\text{c}}}g\sum_{\substack{k\\ \Lambda<|\mathbf{k}|<\Lambda_0}}\left(2G_k+\widetilde{G}_k\right),$$
$$\delta\mu_2 = \sum_{\substack{k\\ \Lambda<|\mathbf{k}|<\Lambda_0}}\left[g\widetilde{G}_k-n_{\text{c}}g^2\left(4G_kG_k+6\widetilde{G}_k\widetilde{G}_k+8G_k\widetilde{G}_k\right)\right],$$
$$\delta\mu_1 = \sum_{\substack{k\\ \Lambda<|\mathbf{k}|<\Lambda_0}}\left[2gG_k-n_{\text{c}}g^2\left(4G_kG_k+2G_kG_{-k}+8G_k\widetilde{G}_k+4\widetilde{G}_k\widetilde{G}_k\right)\right].\quad (39)$$

The change in the coupling $h$ in the linear term (36) through mode elimination corresponds to a depletion of the condensate mode due to zero-point and thermal excitations [1]. This depletion thus appears as a shift of the zero-mode field and is accounted for by splitting the $<$ contribution of the field after mode elimination into a zero-mode contribution with non-vanishing expectation value and one for $k > 0$, with vanishing mean, $\psi_k^< = \delta(k)\delta\sqrt{n_{\text{c}}}+\psi_k'^<$. Inserting this into (8) gives rise to a contribution of the linear term,

$$S_{\text{d}} = \left[\delta h+\delta\sqrt{n_{\text{c}}}(\mu_1+\mu_2)\right]\left(\psi_0'^<+\psi_0'^{*<}\right),\quad (40)$$

neglecting higher-order corrections in the infinitesimal shift $\delta\sqrt{n_{\text{c}}}$. As the action after mode elimination is to be expanded about the new minimum, this linear term must vanish, which fixes the condensate amplitude shift as

$$\delta\sqrt{n_{\text{c}}} = -\delta h/(\mu_1+\mu_2).\quad (41)$$

In the same fashion, the shift in the condensate amplitude gives an additional contribution to the quadratic couplings:

$$\mu_2^< = \mu_2+\delta\mu_2+2\sqrt{n_{\text{c}}}g\,\delta\sqrt{n_{\text{c}}} = \mu_2+\Delta\mu_2,$$
$$\mu_1^< = \mu_1+\delta\mu_1+4\sqrt{n_{\text{c}}}g\,\delta\sqrt{n_{\text{c}}} = \mu_1+\Delta\mu_1.\quad (42)$$

As mentioned earlier, U(1) symmetry requires the two-point couplings $\mu_1$ and $\mu_2$ to be equal, to ensure a gapless quasiparticle mode frequency (32). This relation is a consequence of a U(1) Ward identity [66], which ensures the absence of approximation-related IR divergences in the course of the flow. As a result, the 'hole' dispersion $\omega_k^-$ (33) is gapless. To explicitly demonstrate that the identity $\mu_1 = \mu_2$ holds at one-loop order, we compute the difference between the renormalized couplings using the relation (C.16) between different Matsubara sums:

$$\mu_1^< - \mu_2^< = \mu_1 - \mu_2 + g \sum_{\substack{k \\ \Lambda < |\mathbf{k}| < \Lambda_0}} \left[ 2G_k \left( 1 - \frac{2n_c g}{\mu_1 + \mu_2} \right) + \widetilde{G}_k \left( \frac{2n_c g}{\mu_2} - \frac{2n_c g}{\mu_1 + \mu_2} - 1 \right) \right]. \quad (43)$$

Upon inserting the initial value $\mu_{2,\text{in}} = n_c g$ and using the initial equality of both two-point couplings (9) this expression vanishes, which proves the invariance of $\mu_1 = \mu_2 = \mu$ in the RG flow. The new chemical potential $\mu^<$ after mode elimination thus follows from (42) to be

$$\mu^< = \mu - 2g \sum_{\substack{k \\ \Lambda < |\mathbf{k}| < \Lambda_0}} \left[ G_k + \mu \left( 2G_k G_k + 3\widetilde{G}_k \widetilde{G}_k + 4G_k \widetilde{G}_k \right) \right]. \quad (44)$$

Finally, the renormalized four-point coupling is obtained from the relation $\mu = n_c g$, which is equivalent to the Hugenholtz-Pines relation, i.e., the equality of the couplings, $\mu_1 = \mu_2$, both initialized in (9), which holds throughout the renormalization flow as demonstrated above.

As a result, the new four-point coupling reads

$$g^< = g + \Delta g = g + \frac{\Delta \mu_2}{n_c} - \frac{\mu_2 \delta n_c}{n_c^2} = g + \frac{\delta \mu_2}{n_c}. \quad (45)$$

Hence, while we determined the renormalization of the two-point couplings explicitly, we neglect all newly emerging coupling constants at the order of quartic terms in the field. This will later be relevant when considering the scaling behavior of $g$ which we thus merely infer from the renormalization of the chemical potential.

### 3.2.4 Particle density

Besides the flow equations for both couplings, we need to determine the flow of the total particle density $n = n_c + n_T$, consisting of a condensate and a thermal fraction. The renormalized density is first expanded around the condensate density

$$n^< = \frac{1}{\beta V} \sum_{|\mathbf{k}| < \Lambda} \langle \Psi_k^* \Psi_k \rangle = n_c + \frac{\sqrt{n_c}}{\beta V} \left( \langle \psi_0^< \rangle + \langle \psi_0^{*<} \rangle \right) + \frac{1}{\beta V} \sum_{|\mathbf{k}| < \Lambda} \langle \psi_k^{*<} \psi_k^< \rangle, \quad (46)$$

using $\Psi_k = \delta(k)\sqrt{n_c} + \psi_k^<$. Since the condensate density changes under a mode elimination step, the field $\psi_k^<$ acquires a contribution in the zero mode with non-vanishing expectation value. In order to express (46) in terms of 'thermal' fields $\psi_k'$ with vanishing expectation value, we expand around the change in condensate density $\psi_k^< = \delta(k)\delta\sqrt{n_c} + \psi_k'^<$. This results in the shift of the field expectation value, as by definition $\langle \psi_k'^< \rangle = 0$ in all $k$ modes,

$$\langle \psi_k^< \rangle = \delta(k)\delta\sqrt{n_c} + \langle \psi_k'^< \rangle = \delta(k)\delta\sqrt{n_c}. \quad (47)$$

The change in the condensate amplitude $\delta\sqrt{n_c}$ is obtained from the shift of the coupling $h$ as obtained in (41). Thus, using $\delta(0) = \beta V$, the second term in (46) corresponds to the change in the condensate density. The third term in (46) is expanded to

$$\langle \psi_k^{*<} \psi_k^< \rangle = \langle \psi_k'^{*<} \psi_k'^< \rangle + \mathcal{O}\left( (\delta\sqrt{n_c})^2 \right), \quad (48)$$

where we can neglect the terms of order $(\delta\sqrt{n_c})^2$. As a result, it yields the renormalized density of thermal particles within the remaining range of modes $k < \Lambda$,

$$n_{\mathrm{T}}^< = \frac{1}{\beta V} \sum_{|\mathbf{k}|<\Lambda} \left\langle \psi_k'^{*<} \psi_k'^< \right\rangle. \tag{49}$$

The change of the thermal particle density $\delta n_{\mathrm{T}} = n_{\mathrm{T}}^< - n_{\mathrm{T}}$ can be obtained by performing the momentum shell integration

$$\delta n_{\mathrm{T}} = \frac{1}{\beta V} \sum_{\Lambda<|\mathbf{k}|<\Lambda_0} \left\langle \psi_k^{*>} \psi_k^> \right\rangle. \tag{50}$$

The expectation value $\left\langle \psi_k^{*>} \psi_k^> \right\rangle$ lacks any zero-mode contribution and accounts for an infinitesimal change of the total density. Here, we restrict ourselves to its Gaussian approximation, which is equivalent to the 1-loop approximation we chose in (16),

$$\left\langle \psi_k^{*>} \psi_k^> \right\rangle \simeq \left\langle \psi_k^{*>} \psi_k^> \right\rangle_0 = \delta(0)\,\Theta(|\mathbf{k}|-\Lambda)\,G_k. \tag{51}$$

Inserting the above results into (46) using $n_{\mathrm{T}}^< = n_{\mathrm{T}} + \delta n_{\mathrm{T}}$ and (41), with $\mu_1 = \mu_2$, yields

$$n^< = n_c + 2\sqrt{n_c}\delta\sqrt{n_c} + n_{\mathrm{T}} + \delta n_{\mathrm{T}} = n - \sum_{\substack{k \\ \Lambda<|\mathbf{k}|<\Lambda_0}} \left( G_k + \widetilde{G}_k \right). \tag{52}$$

The momentum integration ranges only from $\Lambda$ to $\Lambda_0$; hence, for

$$l = \ln(b) = \ln(\Lambda_0/\Lambda) \to 0, \tag{53}$$

we obtain the classical expression for the total density.

### 3.2.5 Wave-function renormalization

Wave-function renormalization appears in the symmetry-broken phase already at one-loop order due to the emerging three-point interaction in (10). This gives rise to loop terms in the effective action, of the form (37). As a result, renormalization of the derivative terms in the action appears and gives rise to new terms, at leading order $Z_\tau \partial_\tau$, $V \partial_\tau^2$ and $Z_x \nabla^2$.

In the following, we will account for $Z_\tau \neq 1$ and $V \neq 0$ only and neglect the renormalization of the spatial derivative as well as of derivative terms quadratic in the fields, $\sim \psi^2$ and $\sim \psi^{*2}$. The renormalization of the spatial derivative term, $Z_x \nabla^2$, is expected to remain at the percent level for temperatures sufficiently far away from criticality and thus does not play a role in the quasiparticle regime of sound excitations, where the action shows an approximate $\mathrm{SO}(d+1)$ symmetry [43, 44, 52]. In the vicinity of the critical point, however, renormalization of $Z_x$ would be relevant and thus gives rise to a non-zero anomalous dimension $\eta_x$.

To derive the renormalization of $Z_\tau$ and $V$, we first extract the linear and quadratic contributions in Matsubara frequencies by performing a Taylor expansion of the correlator

$$G_{k'-k} = G_{k'} - \omega_n \dot{G}_{k'} + \frac{\omega_n^2}{2} \ddot{G}_{k'} + \mathcal{O}(\omega_n^3), \tag{54}$$

with frequency derivatives written as

$$\dot{G}_{k'} = \left. \frac{\partial G_k}{\partial \omega_n} \right|_{k=k'}, \qquad \ddot{G}_{k'} = \left. \frac{\partial^2 G_k}{\partial \omega_n^2} \right|_{k=k'}. \tag{55}$$

They are given, together with the corresponding derivatives of $\widetilde{G}_k$, in the appendix, (C.17) and (C.18). In the mode-elimination step, we insert the expansion (54) into (38). $\delta Z_\tau^<$ then follows from the coupling multiplying $-\mathrm{i}\omega_n \psi_k^< \psi_k^{*<}$,

$$\delta Z_\tau^< = 2\mathrm{i}\mu g \sum_{\substack{k \\ \Lambda < |\mathbf{k}| < \Lambda_0}} \left[ 2\left(2G_k + \widetilde{G}_k\right)\dot{\widetilde{G}}_k + \left(2G_k - G_{-k}\right)\dot{G}_k \right], \tag{56}$$

while $\delta V^<$ is read off the term proportional to $\omega_n^2 \psi_k^< \psi_k^{*<}$, cf. (38),

$$\delta V^< = -\mu g \sum_{\substack{k \\ \Lambda < |\mathbf{k}| < \Lambda_0}} \left[ 2\left(2G_k + \widetilde{G}_k\right)\ddot{\widetilde{G}}_k + \left(2G_k + G_{-k}\right)\ddot{G}_k \right]. \tag{57}$$

## 3.3 Rescaling

Typically, within the WRG procedure, the next step is to rescale all coordinates as well as couplings modified by mode elimination, $\mathbf{g}^< = \mathbf{g} + \Delta\mathbf{g}$, by powers of the scale factor $b = \Lambda_0/\Lambda$, which yields the renormalized couplings $\mathbf{g}'$. The goal of this standard procedure, which is outlined, for comparison, in App. B, is to remove all explicit dependencies on the scale factor $b$ and obtain an effective description which, close to criticality, exhibits self-similar scaling behavior.

In order to obtain, however, effective values for the couplings at a fixed length scale – as set by the scattering length – and at a fixed temperature $T$ one rather works without rescaling the coordinates explicitly. This approach allows us to extract bulk quantities like the condensate density in terms of extrinsic units. Thus, only the fields in momentum space are rescaled as

$$\Psi'(\omega_n, \mathbf{k}) = \zeta_b^{-1} \Psi^<(\omega_n, \mathbf{k}), \tag{58}$$

with $\zeta_b$ being the field rescaling factor. Inserting the above scaling relation into $S^<[\mathbf{g}^<]$, cf. (5), and demanding that the action does not rescale leads to the scaling of the wave-function renormalization factors,

$$Z_\tau' = \zeta_b^2 Z_\tau^<, \qquad Z_x' = \zeta_b^2 Z_x^<, \tag{59}$$

where, in this work, we will set and keep $Z_x^< = 1$ throughout.

We furthermore demand $Z_\tau' \overset{!}{=} 1$. As long as $V = 0$, which will be the case as long as the flow has not yet entered the region of scales larger than the healing length (cf. Sect. 4.3.2), this choice ensures the conservation of the canonical commutation relations. Moreover, $Z_\tau' = 1$ ensures the Matsubara frequencies and thus the temperature to remain fixed. With this choice, we read off that

$$\zeta_b^2 = 1/Z_\tau^<, \qquad Z_x' = 1/Z_\tau^<. \tag{60}$$

With these relations, the rescaling of the chemical potential and of the wave-function renormalization factor $V$ result as

$$\mu' = \frac{\mu^<}{Z_\tau^<}, \qquad V' = \frac{V^<}{Z_\tau^<}. \tag{61}$$

Finally, without spatio-temporal rescaling, the total and condensate densities remain invariant,

$$n' = n^<, \qquad n_c' = n_c^<. \tag{62}$$

The rescaling of the interaction coupling,

$$g' = \frac{g^<}{Z_\tau^<}, \tag{63}$$

is then inferred from $\mu = n_c g$ and (61).

From $1 = Z'_\tau = b^{[Z_\tau]} Z^<_\tau$, it follows that the scaling dimension of $Z_\tau$ is given by

$$[Z_\tau] = -\partial_l \ln Z^<_\tau \equiv \eta_\tau \,, \tag{64}$$

expressed in terms of the scale parameter

$$l = \ln(\Lambda_0/\Lambda) = \ln(b) \,. \tag{65}$$

Hence, the emergence of a wave-function renormalization $Z_\tau \neq 1$ in the time derivative leads to the emergence of a temporal anomalous dimension $\eta_\tau$, which is closely related to the dynamical scaling exponent $z = \partial \ln \omega(\mathbf{k})/\partial \ln k$. The latter characterizes the momentum scaling of the quasiparticle frequencies and thus of the density of states, and is in general different from the engineering dimension $[\omega]_{\text{eng}} = -[\tau]_{\text{eng}} = 1$ of frequency as an inverse of time. In the course of the renormalization procedure, within the symmetry-broken phase, $z$ is expected to flow, from the case of free particles ($z = 2$) to sound waves ($z = 1$). In the thermal phase away from criticality, no sound waves appear and $z = 2$ throughout.

The derivative terms in the action imply that, for $Z^<_x \equiv 1$, the kinetic term $\sim \nabla^2$ must scale as the temporal term $Z^<_\tau \partial_\tau$ and thus

$$2 = [Z_\tau] + z = \eta_\tau + z \,, \tag{66}$$

where we used the scaling dimensions of space $[\mathbf{x}] = -1$ and time $[\tau] = -z$, cf. [49]. Along the same lines, also the scaling dimension of $V$ can be related to the dynamical scaling exponent,

$$2 = [V] + 2z \,. \tag{67}$$

In the Klein-Gordon limit, in the IR, the scaling dimension $[V] = -2[c_s]$ relates to the scaling of the sound velocity $c_s$ and hence results in $[c_s] = z - 1$.

## 3.4 Flow equations: Symmetric phase

Combining the mode elimination and rescaling steps and choosing the scale factor $b = \Lambda_0/\Lambda$ to be infinitesimally close to 1, one derives the differential RG equations that govern the flow of the couplings in $l = \ln b$. First of all, the Matsubara sums in (25) and (28) must be evaluated, as described in detail in App. C.1.

Recall that, according to (24), no wave-function renormalization of the two-point couplings appears in the symmetric phase up to one-loop order, which implies that $Z^<_\tau = 1$ and $V^< = 0$ and that the dynamical scaling exponent therefore remains $z = 2$. This will change, however, in the symmetry broken phase described in the next subsection, where $z$ will flow to $z = 1$ in the IR limit.

The free correlator (22) therefore takes the form $G(k) = 1/(i\omega_n + \omega_k)$ with Matsubara frequencies $\omega_n = 2\pi n T$ and the (shifted) free dispersion relation $\omega_k = \varepsilon_k - \mu$, such that the sum in (25),

$$\frac{1}{\beta} \sum_{\omega_n} G(k) = n_B(\omega_k), \tag{68}$$

yields the Bose-Einstein distribution

$$n_B(\omega_k) = \frac{1}{\exp(\beta \omega_k) - 1} \,. \tag{69}$$

In the following we drop the argument, thereby implying that $n_k \equiv n_B(\omega_k)$. The sums in (28) are evaluated in a similar manner, cf. (C.4) in App. C.1. Inserting these results into (25)

and (28), and rescaling according to (61) and (63), one obtains, in the limit $\delta l = 1 - \Lambda/\Lambda_0 \to 0$, the differential flows. Expressed in terms of dimensionless couplings and dispersion,

$$\bar{g} = \frac{S_d \Lambda_0^d}{(2\pi)^d} \frac{g'}{\varepsilon_{\Lambda_0}}, \qquad \bar{\mu} = \frac{\mu'}{\varepsilon_{\Lambda_0}}, \qquad \bar{T} = \frac{T}{\varepsilon_{\Lambda_0}}, \qquad \bar{\omega}_k = \frac{\omega'_k}{\varepsilon_{\Lambda_0}}, \tag{70}$$

with $S_d$ denoting the surface of a $d$-dimensional sphere, the flow equations result as [1]

$$\begin{aligned}
\partial_l \mu &= -2b^{-d} g n_\Lambda, \\
\partial_l g &= -b^{-d} g^2 \left[ 4\beta n_\Lambda(1 + n_\Lambda) + \frac{1 + 2n_\Lambda}{2(b^{-2} - \mu)} \right],
\end{aligned} \tag{71}$$

where we have dropped the overbars for notational simplicity. Note that the dimensionless dispersion in the second term in square brackets is rescaled to $\omega_\Lambda = b^{-2} - \mu$.

## 3.5 Flow equations: Symmetry-broken phase

The flow equations (71) are valid only in the symmetric phase, i.e., for a given density, sufficiently far above the critical temperature, where the field expectation value vanishes and wave-function renormalization of the spatial derivative term can be neglected. In the following, we derive the respective flow equations in the symmetry-broken phase, again sufficiently far away from criticality, keeping $Z_x^< = 1$, but taking into account the flow of $Z_\tau$ and thus the anomalous dimension $\eta_\tau$.

In addition to the dimensionless couplings (70), we need to define the dimensionless densities, quasiparticle frequencies (33) and wave-function renormalization $V$,

$$\begin{aligned}
\bar{n} &= \frac{(2\pi)^d n'}{S_d \Lambda_0^d}, \qquad \bar{n}_c = \frac{(2\pi)^d n'_c}{S_d \Lambda_0^d}, \\
\bar{\omega}_{\Lambda_0}^\pm &= \frac{\omega'^\pm_{\Lambda_0}}{\varepsilon_{\Lambda_0}}, \qquad \bar{V} = \varepsilon_{\Lambda_0} V',
\end{aligned} \tag{72}$$

where $S_d$ denotes again the surface of a $d$-dimensional sphere. Performing the Matsubara sums as summarized in App. C.2, we obtain the flow equations in the symmetry-broken phase for the dimensionless couplings, dropping again the overbars,

$$\begin{aligned}
\partial_l \mu = \ & \eta_\tau \mu - 2b^{-d} g \left[ \left( Z_x b^{-2} + 3\mu \right) K_{1,0} + V K_{1,1} + \mu^2 \left( \mu - 4Z_x b^{-2} \right) K_{2,0} \right. \\
& \left. - 4\mu(1 + \mu V) K_{2,1} - 1 \right], \\
\partial_l g = \ & \eta_\tau g - b^{-d} g^2 \left[ 5K_{1,0} + 2\mu \left( \mu - 4Z_x b^{-2} \right) K_{2,0} - 8(1 + \mu V) K_{2,1} \right],
\end{aligned} \tag{73}$$

and densities,

$$\begin{aligned}
\partial_l n &= b^{-d} \left( -Z_x b^{-2} K_{1,0} - V K_{1,1} + 1 \right), \\
\partial_l n_c &= b^{-d} \left[ -\left( 2Z_x b^{-2} + \mu \right) K_{1,0} - 2V K_{1,1} + 2 \right].
\end{aligned} \tag{74}$$

The auxiliary functions $K_{i,j}$ are defined in App. C.2 in their dimensionless form. The flow equation for the condensate density can be obtained from $n_c(l) = \mu(l)/g(l)$.

Note that in the limit of vanishing wave-function renormalization, $Z_\tau^< \equiv 1$ and $V^< \equiv 0$, and thus $z = 2$ and $\eta_\tau = 0$, the flow equations (73) and (74) reduce to those derived in [1].

Extending upon this description, we find, from (56), the flow equation for $Z_\tau^<(l)$ and, using (57), the flow of $V$. With the Matsubara sums given in (C.19) and (C.20) of App. C.2,

expressed in terms of rescaled and dimensionless couplings, the temporal anomalous dimension $\eta_\tau = -\partial_l \ln Z_\tau^<$ results as

$$\eta_\tau = 2b^{-d} Z_x \mu g \left[ \left( Z_x b^{-2} + \mu \right) K_{2,0} - 9V K_{2,1} + 8 Z_x (2\mu V - 1) K_{3,1} + 8V(2\mu V + 1) K_{3,2} \right]. \quad (75)$$

As can be seen from Eqs. (73) and (74), it affects the flows of both, the couplings and the densities.

The flow of the coupling $V$ is determined analogously, using the Matsubara sums (C.21) and (C.22), and results as

$$\begin{aligned}
\partial_l V = \; & \eta_\tau V - b^{-d} Z_x \mu g \Big\{ -3(1+A_k) K_{2,0} + 18V^2 K_{2,1} + 2\mu A_k \left( 4 Z_x b^{-2} - \mu \right) K_{3,0} \\
& - 4 \left[ 2 + V^2 \left( 4 Z_x^2 b^{-4} + 28 Z_x b^{-2} \mu - 13\mu^2 \right) - A_k \left( 11 + 2V\omega^2 \right) \right] K_{3,1} \\
& - 16 V^2 (3 + 5\mu V) K_{3,2} + 8 \left( 4V^2 \omega^2 - A_k^2 \right) \\
& \times \left[ \mu \left( 4 Z_x b^{-2} - \mu \right) K_{4,1} + 4(1 + \mu V) K_{4,2} \right] \Big\},
\end{aligned} \quad (76)$$

with $A_k = 1 + 2V \left( Z_x b^{-2} + \mu \right)$ in its rescaled form. The flow equations (73)–(76) constitute a closed set of equations defining the RG evolution including wave-function renormalization of the first- and second-order temporal derivative terms.

# 4 Solving the flow equations

## 4.1 Initialization of the flow equations

We can now use the flow equations (73) and (74) to determine the evolution of the couplings in the condensed phase. Before doing so, we need to initialize them at the UV cut-off $\Lambda_0$. We start with the quartic coupling and thereafter define the initialization of the remaining quantities.

### 4.1.1 Initialization of the coupling at a chosen microscopic scale

Solving the flow equations (73) and (74), starting at the scale set by the UV cut-off $\Lambda_0$, yields solutions, which in principle depend on this initial scale $\Lambda_0$. However, the cut-off scale should not have any physical significance for the IR physics since it merely defines the limit of validity of the $s$-wave approximation. A cut-off independent flow in the IR can be achieved by considering the experimentally measured scattering length $a_0$ as a renormalized quantity measured in vacuum. This is reasonable since the experimental value is already an effective quantity containing quantum corrections. Hence, we initialize the flow such that, in vacuum, the flowing scattering length eventually reaches the experimental value of $a_0$ [1]. The vacuum flow equations can be obtained from those in the symmetric phase, Eqs. (71), by setting $n_k = 0$. Assuming a small chemical potential, $\mu \ll 1$, the equation can be analytically solved to yield

$$g(l) = \frac{2\epsilon g_{\text{in}}}{2\epsilon - g_{\text{in}} + e^{\epsilon l} g_{\text{in}}}, \quad (77)$$

where we introduced the scaling dimension $\epsilon = 2 - d$ for the interaction coupling. We determine, from this, the bare coupling $g_{\text{in}}$ in $d = 3$ dimensions, where $\epsilon = -1$, since no wave-function renormalization is present in the thermal phase. The dimensionless four-point coupling, cf. (70), in vacuum asymptotically approaches the $s$-wave result

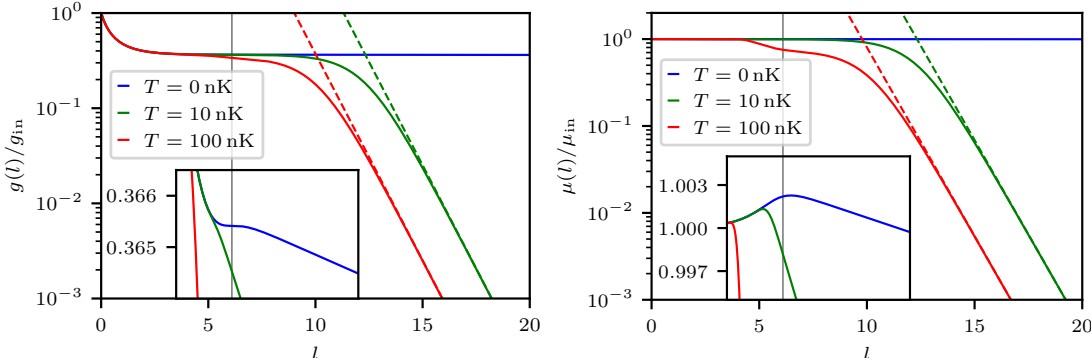

Figure 1: RG flow of the quartic coupling $g(l)$ (left) and the chemical potential $\mu(l)$ (right) normalized by their initial values $\mu_{\text{in}}$ and $g_{\text{in}}$, respectively. The couplings are shown as functions of the flow parameter $l = \ln(\Lambda_0/\Lambda)$ encoding the flowing cut-off scale $\Lambda$, for three different temperatures below the non-interacting critical temperature $T_c^0 \approx 324\,\text{nK}$ of Bose-Einstein condensation in a 3-dimensional gas of $^{23}$Na atoms of density $n = 10^{19}\text{m}^{-3}$. No wave-function renormalization has been taken into account here. The initial UV cut-off is chosen to be $\Lambda_0 = 1/a_0$, with the $s$-wave scattering length $a_0$. The healing-length scale $l_\xi = \ln(\xi \Lambda_0)$ is marked by the gray line. The insets show the flow in the vicinity of the healing-length scale on an enlarged $y$-scale and with the $x$-axis shared with the main plots. In both graphs, the dashed lines represent the asymptotic solutions $g^*(l) = 2b^{-1}\beta$ and $\mu^*(l) = n_{\text{c,phys}}g^*(l)$, respectively, which exhibit anomalous scaling dimensions $[\mu]_{\text{scaling}} = 1$ and $[g]_{\text{scaling}} = -2$ compared to the engineering dimensions $[\mu]_{\text{eng}} = 2$ and $[g]_{\text{eng}} = -1$.

$g(l \gg 1) = (m\Lambda_0/\pi^2) \times (4\pi a_0/m) = 4\Lambda_0 a_0/\pi$ for large flow parameter $l = \ln(b) = \ln(\Lambda_0/\Lambda)$. Thus, the initial value for $g$ from (77) is

$$g_{\text{in}} = \frac{4a_0\Lambda_0}{\pi - 2a_0\Lambda_0}\,. \tag{78}$$

We will employ this initial, bare coupling in the non-vacuum flow when determining the low-energy effective four-point coupling in the macroscopic regime. By making, in this way, the initial value cut-off dependent we achieve cut-off independent outcomes and fix the flow equations to a particular physical vacuum scattering length.

### 4.1.2 Initialization of the chemical potential, temperature, and density

We begin solving the flow equations by initializing all particles in the symmetry-broken phase, $n_{\text{in}} = \mu_{\text{in}}/g_{\text{in}}$. The initial chemical potential $\mu_{\text{in}}$ is determined by means of the UV cut-off dependent initial coupling $g_{\text{in}}$ (78) and the set initial density. We set the UV cut-off to $\Lambda_0 = \gamma/a_0$, with the free parameter $\gamma$ set to $\gamma = 1$ unless stated otherwise. The temperature $T$ is set to the physical temperature.

For a fixed initial total particle density $n_{\text{in}}$, the density in the IR would depend on the chosen temperature. Therefore, to obtain a desired physical density $n(l \to \infty) = n_{\text{phys}}$, one needs to correctly specify the initial condition $n(l = 0) = n_{\text{in}}(T)$ for each given temperature $T$. Thus, the initial density $n_{\text{in}}$ can be found as the root of $n(n_{\text{in}}, l \to \infty) - n_{\text{phys}}$ using the secant method. The initial condensate density is set to the initial total particle density, i.e., all particles are initialized in the condensate.

During the RG flow, the chemical potential will remain positive but will decrease due to the redistribution of condensed particles into the excited states. For temperatures at and above $T_c$, the chemical potential vanishes at some point during the flow. This moment indicates the

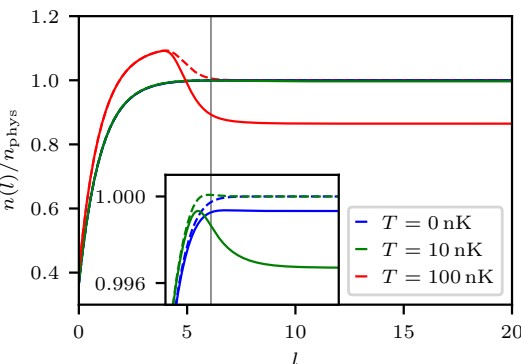

Figure 2: RG flow of the condensate density $n_c(l)$ (solid) and the total particle density $n_{tot}(l)$ (dashed) normalized by the final density $n_{phys} = 10^{19} \text{m}^{-3}$. The densities are plotted against the flow parameter $l$ without wave-function renormalization below the critical temperature. As before, the UV cut-off is $\Lambda_0 = 1/a_0$ and the healing length scale is indicated by the gray line. The subplot, which shares the $x$-axis with the main plot, shows the flow around the healing scale enlarged and exhibits a condensate depletion at $T = 0$. For large $l$, i.e., in the IR limit, one observes convergence to constant values.

restoration of the thermal phase as no particles are condensed anymore. Beyond this point, the couplings evolve according to the thermal flow equations (71). A smooth transition between the phases is guaranteed since the flow equations for both couplings are identical for $\mu = 0$.

## 4.2 Numerical results without wave-function renormalization

Given the initial values for the couplings and the densities, the flow equations are numerically solved and the renormalized couplings are read out at some finite maximum value $l_{max}$. In the remainder of this subsection, we neglect wave-function renormalization by setting the respective factors to $Z_\tau^< = 1$ and $V = 0$ and demonstrate how this leads to a diverging flow of physical quantities and the necessity to take wave-function renormalization into account in the IR regime where collective quasiparticle excitations prevail.

### 4.2.1 Flows of the couplings and densities

In order to discuss the RG flows at a chosen momentum and a temperature scale, we work within the rescaling scheme defined in Eqs. (61)–(63). Since $Z_\tau^< = 1$ and $V = 0$, this means that no rescaling is applied, $\mu'(l) = \mu^<(l)$, $g'(l) = g^<(l)$, and $n'(l) = n^<(l)$.

All the numerical evaluations are performed for parameters describing the example of a dilute Bose gas of $^{23}$Na atoms with the $s$-wave scattering length $a_0 = 51.12\,a_B$ in the $(F, m_F) + (F, m_F) = (1,0) + (1,0)$ hyperfine channel [67][1] and a particle density $n_{phys} = 10^{19}/\text{m}^3$. All RG flows were evolved from a UV cut-off $\Lambda_0 = 1/a_0$ up to a maximum flow parameter $l_{max} = 20$ situated deep in the IR.

Fig. 1 shows the flows of the chemical potential $\mu(l)$ and the interaction coupling $g(l)$, both normalized to their respective initial values. The vertical gray line indicates the healing-length scale, $\Lambda_\xi^{-1} = e^{l_\xi}/\Lambda_0 = \xi = (8\pi a_0 n_{phys})^{-1/2}$ which has been shown to be the relevant length scale for the transition from a quadratic towards a linear dispersion relation [49]. For the chosen density and scattering length, we obtain a healing-length scale parameter $l_\xi \approx 6.1$.

---

[1]See [68] for more recent data on scattering lengths, which would, however, not change our results here.

Sufficiently far below this scale one observes qualitatively similar flows for all three temperatures chosen, $T = 0$, 10, and 100 nK. Beyond the healing-length scale, the flows run into asymptotic solutions which exhibit scaling in $b = e^l$. As a result, there is no convergence even for $T = 0$, for which the scaling is strongly suppressed, see the insets.

Besides the healing length scale, the Ginzburg momentum $k_G$ [35] indicates the scale at which the perturbative approximation breaks down. A perturbative approximation of $k_G$ has been put forward in [35, 47]. However, since the crossover from a linear to a quadratic dispersion relation is governed by the healing-length scale $k_\xi$ [49] and an extension featuring a convergence of $V$ is beyond our scope here, we do not consider the Ginzburg in more detail here.

Fig. 2 shows the flows of $n_{tot}(l)$ (dashed) and $n_c(l)$ (solid lines) normalized to the chosen asymptotic total density $n_{phys} = 10^{19} m^{-3}$, for the same three temperatures as above. As anticipated, the condensate density decreases with increasing temperature, whereas the total density flows towards the chosen final total density. In contrast to the couplings, we observe convergence in the flows of both densities. As a result, the asymptotic behavior of the condensate and total densities results as $n_c^*(l) = n_{c,phys}$ and $n^*(l) = n_{phys}$, respectively. The $T$-dependence of $n_{c,phys}$ will be examined in more details below.

### 4.2.2 Asymptotic scaling

The observed asymptotic scaling behavior, i.e. for $l \gg 1$, for non-zero temperatures, can be read off the approximate asymptotic forms of the flow equations (73) and (74) as the leading order contribution in powers of $b$. Using the auxiliary functions defined in App. C.2 and the asymptotic dispersion relation $\omega_k^2 \to 2\mu/b^2$ and Bose-Einstein distribution $n_k \to 1/(\beta\omega_k)$ leads to the following asymptotic flow equations:

$$\partial_l g = -b^{4-d} \frac{g^2}{2\beta}, \qquad \partial_l \mu = -b^{4-d} \frac{g\mu}{2\beta},$$
$$\partial_l n = 0, \qquad \partial_l n_c = 0. \tag{79}$$

For $n$ and $n_c$, the flows converge as found in Fig. 2. The coupling approaches the asymptotic scaling behavior $g(l) \sim g^*(l) = 2b^{-1}\beta$ satisfying $\partial_l b g^*(l) = b(1 + \partial_l)g^*(l) = 0$, as can be inferred from the flow equation (79) for the coupling $g$ in $d = 3$ dimensions. This behavior is reflected by the exponential decay for $l \gtrsim 15$ in the left panel of Fig. 1, depicting the normalized $g(l)/g_{in}$. For comparison, we also show, as dashed lines, the respective asymptotic scaling $2b^{-1}\beta/g_{in}$.

Analogously, the flow equation for $\mu$ gives the asymptotic scaling $\mu(l) \sim \mu^*(l) \sim b^{-1}$ at large $l$, as can be seen in the right panel of Fig. 1. Combining the asymptotic scaling of the coupling and the density confirms that the chemical potential scales as $\mu^*(l) = n_c^*(l)g^*(l) = 2b^{-1}n_{c,phys}\beta$.

If the couplings were scaling with their engineering dimensions, $[\mu]_{eng} = 2$ and $[g]_{eng} = 2 - d$, cf. App. B, then $\mu(l)$ and $g(l)$ would approach constants asymptotically. However, both scale as $b^{-1}$, exhibiting scaling dimensions $[\mu]_{scaling} = [\mu]_{eng} - 1 = 1$ and $[g]_{scaling} = [g]_{eng} - 1 = -2$ in three dimensions. This also implies the absence of cut-off independence, as exhibited by the diverging flows in Fig. 1. Since both couplings deviate in their observed scaling by the same single power in $b$, the engineering and scaling dimensions of the density agree with each other, $[n]_{scaling} = [n]_{eng} = d = 3$.

While anomalous scaling, i.e., a deviation between engineering and scaling dimensions is a generic feature of RG flows close to critical points, it is in general not expected to occur away from criticality, in particular close to zero temperature. The appearance of anomalous scaling is caused by the renormalization of operators such as $\nabla^2$ and $\partial_\tau$ which emerges from

the Taylor expansion of the two-point diagram (37), cf. the discussion in [1]. To be able to avoid taking into account the flow of $Z_\tau$ and $Z_x$, one needs to restrict the description to the regime $na_0 \Lambda_{th}^2 \ll 1$, in which the thermal energy is much larger than the chemical potential and thus the linear part of the Bogoliubov dispersion becomes negligible and IR divergent flows are absent. At more macroscopic scales, in the sound-wave regime, though, one needs to go beyond this approximation. This regime, in the RG framework, so far has been investigated by means of non-perturbative, functional approaches, cf. Refs. [43,44,69]. There, flow equations have been derived, in which the second-order temporal derivative $\partial_\tau^2$ becomes dominant as compared with the first-order derivative $\partial_\tau$ [69]. In the following, we will demonstrate how this transition can be qualitatively described also at the perturbative level.

## 4.3 Numerical results including wave-function renormalization

In the following, we derive the flow equations taking into account the wave-function renormalization factors $Z_\tau$ and $V$. Following the same procedure as without this renormalization, the system is prepared in the fully condensed state, where UV cut-off dependent initial values for the couplings are chosen and the density flows to a pre-determined value of the physical density $n_{phys}$.

### 4.3.1 Asymptotic scaling

We start by analyzing the asymptotic flow equations in order to determine the significance of the wave-function renormalization for the asymptotic scaling. Including now the anomalous dimension $\eta_\tau$, the leading terms in the flow equations (73) for the couplings $\mu$ and $g$ read:

$$\partial_l g = \eta_\tau g - b^{4-d} \frac{Z_\tau g^2}{2\beta}, \qquad \partial_l \mu = \eta_\tau \mu - b^{4-d} \frac{Z_\tau \mu g}{2\beta}. \tag{80}$$

The asymptotic anomalous dimension is given by

$$\eta_\tau = b^{4-d} \frac{Z_\tau g}{2\beta}, \tag{81}$$

cf. (75). Thus, it follows from Eqs. (80) that

$$\partial_l g = 0, \qquad \partial_l \mu = 0, \tag{82}$$

asymptotically. Hence, the second-order one-loop contribution to the changes arising from mode-elimination decreases the scaling dimension of both couplings by the temporal anomalous dimension. This shows that the rescaled couplings $g'(l) = g^<(l)/Z_\tau^<(l)$ and $\mu'(l) = \mu^<(l)/Z_\tau^<(l)$, as defined in (61) and (63), each have a converging flow, cf. (82).

The asymptotic flow equation for $g$, Eq. (82), has the solution $g(l) = g^*$ with $g^*$ being the fixed-point value of the coupling. The flow of the scale-dependent anomalous dimension, cf. (64), is therefore determined by the ordinary differential equation

$$\partial_l \eta_\tau = (4 - d - \eta_\tau)\, \eta_\tau, \tag{83}$$

as follows from (81) using $\partial_l g = 0$. It possesses a fixed point at $\eta_\tau^* = 0$, which corresponds to the free gas with dynamical scaling exponent $z^* = 2$, cf. (66). The second, non-vanishing fixed-point of the anomalous dimension depends on the dimensionality,

$$\eta_\tau^* = 4 - d. \tag{84}$$

Thus, in $d = 3$ dimensions, the anomalous dimension flows to $\eta_\tau(l) \to 1$, which, according to (66), corresponds to $z(l) \to 1$, as it is expected for sound waves.

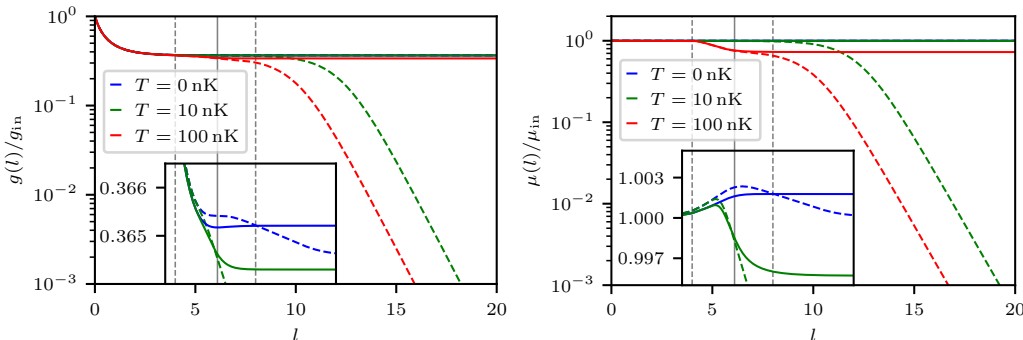

Figure 3: Quartic coupling $g(l)$ (left) and chemical potential $\mu(l)$ (right) normalized by their initial values $\mu_{\text{in}}$ and $g_{\text{in}}$, respectively. The couplings are plotted against the flow parameter $l$ including wave-function renormalization within the symmetry-broken phase. Solid lines correspond to rescaled couplings according to (63) and (61) and show IR convergent flows, whereas dashed lines correspond to the un-rescaled couplings $g^<(l)$ and $\mu^<(l)$. The solid, vertical, gray line indicates again the healing scale, whereas the two dashed lines separate the three distinct intervals of the RG flows, see the discussion in the main text. The insets, which share the $x$-axis with the main plots, show the flow on an enlarged $y$-scale.

### 4.3.2  Flows of the couplings and densities

When numerically solving the flow equations we need to take care of singularities appearing due to the additional coupling $V$. The derivative couplings are initialized as $Z_{\tau,\text{in}} = 1$ and $V_{\text{in}} = 0$. In this limit, the "anti-particle" modes formally decouple: $\omega_k^+ \to \infty$. For $V > 0$, there are two additional modes, which, for $\mu_1 = \mu_2 = m + \mu_{\text{NR}}$, with rest mass $m$ and non-relativistic chemical potential $\mu_{\text{NR}} \simeq gn$, reflect the emergent SO$(d+1)$ symmetry of the model in the sound-wave regime. These modes account for anti-particles, which are equivalent to the particles due to the gapless dispersion. Note that, while the description is similar to that of relativistic massless bosons, we here describe the transition from thermal energies in the single-particle limit to the phononic regime of collective excitations and therefore choose $\mu_1 = \mu_2 = \mu$.

Fig. 3 shows the flows of the quartic coupling $g(l)$ and the chemical potential $\mu(l)$, for the same three temperatures as in Fig. 1. While the unrescaled (dashed) couplings $\mu^<(l)$ and $g^<(l)$ vanish asymptotically at large $l$, as was the case without wave-function renormalization, they reach a constant value once rescaled with $Z_\tau^<(l)$ (solid). As discussed in Sect. 3.3, this rescaling corresponds to fixing the spatial and temporal scales to the experimental scattering length and the physical temperature, respectively. Within this rescaling scheme, only the fields need to be rescaled by $1/Z_\tau^<$, cf. (61) and (63).

In Fig. 4, we display the wave-function renormalization $Z_\tau^<(l)$ (solid), the dynamical scaling exponent $z(l)$ (dashed), and the temporal anomalous dimension $\eta_\tau(l)$ (dotted). For $T \neq 0$, they demonstrate the predicted behavior (84). Beyond the healing scale, the transition into the quasi-relativistic regime leads to a decrease of $Z_\tau^<(l)$, which eventually asymptotically vanishes as $Z_\tau^<(l) \sim 1/b$. Meanwhile, the second-order temporal derivative term grows up leading to a quasi-relativistic form of the action. The temporal anomalous dimension reaches the expected limit $\eta_\tau(l) \to 1$, see (84). Since the dynamical scaling exponent is directly related to the wave-function renormalization via the temporal anomalous dimension (66), one finds $z(l) \to 1$ (cf. Fig. 4). While we could reproduce the above scaling of the wave-function renormalization $Z_\tau^<(l)$ down to $T = 0.1$nK, we find, in the case of $T = 0$, that $Z_\tau^<(l)$ reaches a fixed-point value at $Z_\tau^{<*} \approx 0.9984$ as the temporal anomalous dimension $\eta_\tau$ stays at zero.

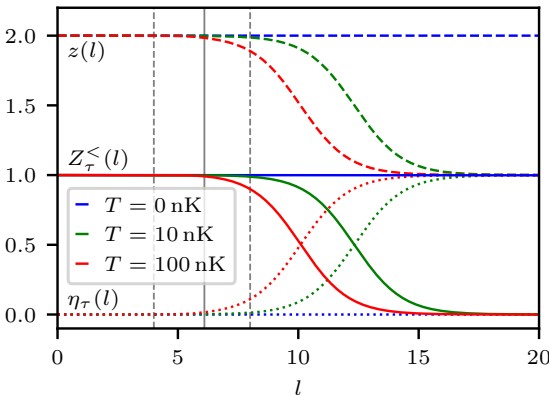

Figure 4: The wave-function renormalization $Z_\tau^<(l)$ (solid), the dynamical scaling exponent $z(l)$ (dashed), and the temporal anomalous dimension $\eta_\tau(l)$ (dotted) plotted against the flow parameter $l$. In the case of non-zero temperature, the transition towards a relativistic regime with $Z_\tau^<(l) \to 0$, $z(l) \to 1$, and $\eta_\tau(l) \to 1$ beyond the healing scale is observed. At $T = 0$, no such transition occurs.

We finally discuss the relevance of the flow of the wave-function renormalization factor $V$. During an initial period of the flow, as long as $l \lesssim l_1 < l_\xi$, the value of $V$ remains sufficiently small such that it does not account for a significant change of the right-hand sides of the flow equations compared to $V = 0$. This, however, changes once the flow reaches the healing-length scale $l \approx l_\xi \approx 6$, cf. the discussion in 4.2.1, where $V(l)$ becomes relevant [44]. Finally, in the asymptotic limit $l \to \infty$, the coupling $V$ decouples again from the flow equations, as can be inferred from the explicit forms of the functions $K_{i,j}$ quoted in App. C.2.

The behavior of the wave-function renormalization factor $V$ as described above leads to numerical difficulties when evolving the flow equations. For small flow parameter $l \ll l_\xi$ the dispersion relation $\omega_k^+ \to \infty$ (33) is effectively decoupled, thus, leading to numerical inaccuracies. At large flow parameters $l \gg l_\xi$ the growth of $V$ leads to a numerically slow solution, even though its effect on the flow equations is negligible.

For these reasons, when numerically solving the flow equations we employ different schemes depending on the value of the flow parameter $l$. At small values $l \leq l_1$, we evolve the flow equations with the quadratic derivative term left out, i.e., $V = 0$, while we compute the growth of $V$ using (76). In the intermediate region $l_1 < l \leq l_2$, we evolve the full flow equations including the feedback of $V$. Finally, at large $l > l_2$, the couplings are evolved using the asymptotic flow equations for $V \gg 1$, obtained from (73) and (75), which ultimately approach (80) and (81), respectively. The values of $l_1$ and $l_2$ are chosen such that a smooth transition is warranted, i.e., that the effect of $V \neq 0$ is small and the feedback of $V$ on the flow equations becomes negligible. In our computations, we chose $l_1 = 4$ and $l_2 = 8$, which we indicate by the gray dashed lines in Figs. 3 and 4. The flow of $V$ is shown in Fig. 7 in App. D.

### 4.3.3 Temperature dependence of the couplings

Having achieved converging RG flows we can investigate the temperature-dependence of various physical quantities. The left panel of Fig. 5 depicts the chemical potential $\mu$ (dashed) and the quartic coupling $g$ (dotted) normalized to their zero-temperature mean-field values $\mu_{MF}$ and $g_{MF}$, respectively, as functions of temperature. One observes a phase transition at around $T_c \approx 340\,\text{nK}$. As expected, the presence of interactions shifts the critical temperature to a larger value compared to the non-interacting result $T_c^0 \approx 324\,\text{nK}$. This shift does not change qualitatively when setting $V = 0$, cf. App. D. The interaction coupling $g$ remains close to its

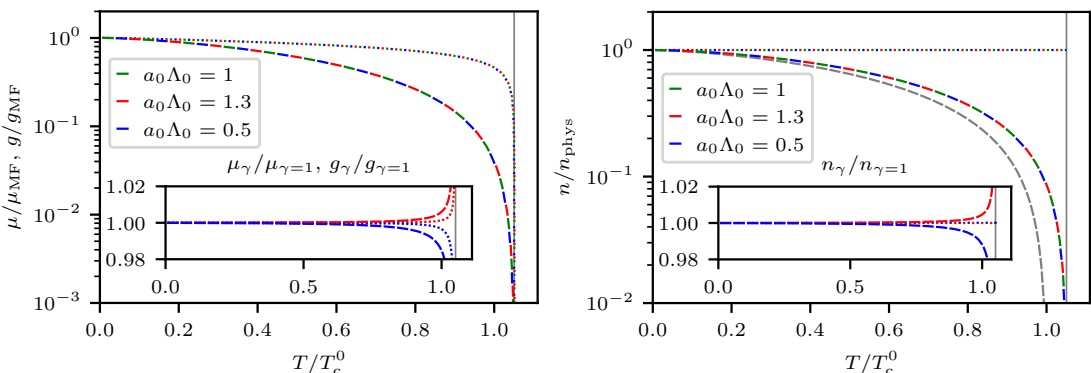

Figure 5: (Left panel) The chemical potential $\mu$ (dashed) and the quartic coupling $g$ (dotted) normalized by their respective zero-$T$ mean-field value $\mu_{\mathrm{MF}}$ and $g_{\mathrm{MF}}$ plotted against temperature $T$. Both couplings are rescaled according to (61) and evaluated at the maximal flow parameter $l_{\mathrm{max}} = 20$. The couplings are plotted for three different UV cut-offs and show no cut-off dependence, which is reflected in the alternating color scheme indicating overlapping results. The inset shows the relative deviation of the $\gamma = a_0 \Lambda_0 = 1.3$ (red) and $\gamma = 0.5$ (blue) results from the $\gamma = 1$ case for both couplings. (Right panel) The condensate density $n_{\mathrm{c}}$ (dashed) and the total density $n_{\mathrm{tot}}$ (dotted) normalized by the physical particle density $n_{\mathrm{phys}} = 10^{19}\,\mathrm{m}^{-3}$ plotted against temperature $T$ for three different values of the UV cut-off. The densities are rescaled according to (62) and evaluated at the maximal flow parameter $l_{\mathrm{max}} = 20$. The condensate density of the non-interacting result (dashed, gray) exhibits the shift to a larger critical temperature. The inset shows the relative deviation of the $\gamma = a_0 \Lambda_0 = 1.3$ (red) and $\gamma = 0.5$ (blue) results from the $\gamma = 1$ case for both densities. The vertical gray line indicates the critical temperature.

mean-field value for most temperatures and falls significantly only in the vicinity of the critical temperature. Thus, the steady decrease of the chemical potential stems from the depletion of the condensate density. As can be seen in the right panel of Fig. 5, this decrease of the condensate density (dashed) with growing temperature is qualitatively similar to that of the non-interacting gas (dashed, gray). The dotted horizontal line demonstrates that we chose the total density constant over the whole temperature range.

Fig. 5 furthermore confirms the desired weak dependence of the chemical potential, coupling, and densities as functions of temperature on the flowing cut-off scale. This is reflected in the alternating color scheme indicating overlapping results for various UV cut-offs. The relative deviations of the $\gamma = a_0 \Lambda_0 = 1.3$ and $\gamma = 0.5$ functions from those at $\gamma = 1$ are shown in the insets and remain below 1% for all temperatures sufficiently far away from the critical point. This is the result of both, the UV cut-off independent initialization described in Sect. 4.1.1 and of the procedure, which ensures convergence, cf. Eqs. (61) and (63). Hence, including wave-function renormalization allows us making quantitative predictions in the long-wave-length regime $\Lambda_{\mathrm{th}}^2 \ll (n a_0)^{-1}$, where the excitations have phononic character.

To benchmark our approach, we determine the condensate depletion from the flow as depicted in Fig. 2. At zero temperature the flow equations for the condensate density are solved for a range of total densities $n_{\mathrm{tot}}$. The results are shown, as functions of the diluteness parameter $\eta = \sqrt{n_{\mathrm{tot}} a_0^3}$, in the left panel of Fig. 6, where we observe good agreement with the leading-order Bogoliubov result

$$1 - \frac{n_{\mathrm{c}}}{n_{\mathrm{tot}}} = \frac{8}{3\sqrt{\pi}} \sqrt{n_{\mathrm{tot}} a_0^3}. \tag{85}$$

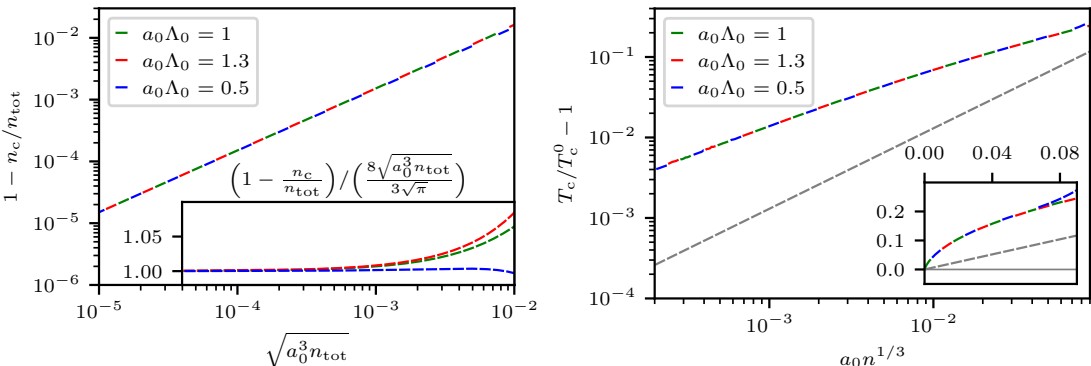

Figure 6: (Left panel) The condensate depletion $1 - n_c/n_{tot}$ for three different values of the UV cut-off evaluated at $l_{max} = 20$ and rescaled according to (62). The alternating color scheme indicates overlapping results and thus cut-off independence. The subplot, which shares the $x$-axis with the main plot, displays the relative deviation from the Bogoliubov prediction (85) showing a good agreement over many orders of magnitude. (Right panel) The critical temperature $T_c$ normalized by the non-interacting critical temperature $T_c^0$ for three different values of the UV cut-off. The alternating color scheme indicates again overlapping results and thus cut-off independence within most of the range of couplings and densities shown. We observe a shift to larger critical temperatures as expected but the linear dependence on $a_0 n^{1/3}$ predicted as $T_c/T_c^0 - 1 = \kappa a_0 n^{1/3}$, with $\kappa = 1.3$ (gray, dashed), appears only for much smaller $a_0 n^{1/3}$, with a larger prefactor $\kappa \approx 19$ (as also seen in the inset on a linear scale).

The relative deviation from this result, shown in the inset, demonstrates corrections on the order of a few percent for large densities. Furthermore, the deviation is not cut-off independent anymore showcasing limitations of our initialization scheme that assumes small chemical potential, see the discussion in Sect. 4.1.

We finally consider the interaction-induced shift of the critical temperature $\Delta T_c/T_c^0 = (T_c - T_c^0)/T_c^0$ relative to the ideal-gas value $T_c^0 = 2\pi/m \, (n/\zeta(3/2))^{2/3}$, with $\zeta(3/2) \approx 2.61$ being the Riemann zeta function. This relative shift is known to obey $\Delta T_c/T_c^0 = \kappa a_0 n^{1/3}$ at leading order $\eta^{2/3}$ in the diluteness parameter. A proportionality constant of $\kappa \approx 1.3$ has been found in different types of Monte-Carlo simulations, from variational perturbation theory, and functional renormalization group theory, see [9] for a review, as well as [1, 29, 36–41, 44, 45, 50, 53–58]. For an RG calculation on the basis of the thermal phase flow equations, cf. (71), together with non-perturbative input see [39]. Here we compute critical temperatures from the symmetry-broken flow equations (73) for a range of different densities as shown in the right panel of Fig. 6. At small densities, the critical temperature approaches its non-interacting value $T_c^0$. Increasing the density leads to a shift to larger critical temperatures in accordance with known results [9]. In terms of the critical degeneracy parameter $n\Lambda_{th}^3$ our results agree with those in [1]. The linear dependence of the shift on $a_0 n^{1/3}$ is only approached for small diluteness parameter with $\kappa$ being an order of magnitude larger than the standard result, as can be seen in the right panel of Fig. 6. This deviation can be attributed to the perturbative nature of the approach employed here. Including, e.g., in an fRG approach, close to the phase transition, the single-particle propagator over a sufficiently large range of momenta yields a $\kappa$ constant compatible with Monte Carlo results [50].

## 5 Conclusions

We applied perturbative Wilsonian renormalization-group theory to computing observables in the symmetry-broken, condensate phase of a weakly-interacting Bose gas in three spatial dimensions. Extending upon the approach of [1] by taking into account, to one-loop order, wave-function renormalization of both the first- and second-order derivative terms in the effective action with respect to imaginary time, we could follow the system into the regime where sound-wave collective quasiparticle excitations prevail. Without this wave-function renormalization included, the flows exhibited unphysical scalings of certain couplings, which thus vanished at wave lengths beyond the healing length scale. Including temporal wave-function renormalization, in contrast, and adopting a rescaling scheme which fixes both the spatial and temporal scales, we were able to achieve converging flows. This allowed us to describe the transition of the system from the particle-like into the phononic regime. Hence, we found the dynamical scaling exponent flowing towards $z \to 1$ marking the emergence of a linear sound dispersion in line with the non-perturbative Bogoliubov-de Gennes description. We used the RG flow equations to compute the depletion of the condensate density due to thermal fluctuations and determine the shift of the critical temperature. We found the depletion to be in good agreement with Bogoliubov-de Gennes theory up to linear order in the diluteness parameter $\eta = \sqrt{na^3}$. For the critical temperature a positive shift was found, which, however, did not linearly scale in $a_0 n^{1/3}$. A further improvement to the perturbative Wilsonian approach adopted here requires including the spatial wave-function renormalization, which is expected to play a significant role close to the critical point. Comparisons with other approaches, in particular with schemes and truncations chosen within the realm of functional RG, as cited above, should be of interest in future work for clarifying the consistency of approximations available to date. Our approach, moreover, can be considered as a first step in extending perturbative RG treatments to describe non-universal properties also in multicomponent Bose gases.

## Acknowledgments

The authors thank A. Baum, P. Heinen, M. K. Oberthaler, H. Köper, J. Mayr, J. M. Pawlowski, A. Pelster, I. Siovitz, and H. T. C. Stoof for discussions and collaboration on related topics.

**Funding information** The authors acknowledge support by the German Research Foundation (DFG), through SFB 1225 ISOQUANT (Project-ID 273811115), grant GA677/10-1, and under Germany's Excellence Strategy – EXC 2181/1 – 390900948 (the Heidelberg STRUCTURES Excellence Cluster). N. R. acknowledges support from the Studienstiftung des deutschen Volkes. A. N. M. acknowledges financial support by the IMPRS-QD (International Max Planck Research School for Quantum Dynamics).

## A  Fourier transformations

Fourier transforms in the Euclidean $d + 1$-dimensional space-time are defined as

$$
\Psi(\tau, \mathbf{x}) = \frac{1}{\beta} \sum_{\omega_n} \int \frac{d\mathbf{k}}{(2\pi)^d} \Psi(\omega_n, \mathbf{k}) e^{-i(\omega_n \tau + \mathbf{kx})} ,
$$
$$
\Psi(\omega_n, \mathbf{k}) = \int_0^\beta d\tau \int d\mathbf{x} \, \Psi(\tau, \mathbf{x}) e^{+i(\omega_n \tau + \mathbf{kx})} .
$$
(A.1)

# B Rescaling including coordinates

In this appendix, we outline the standard rescaling procedure, which can be used to discuss critical exponents in the vicinity of fixed-points of the RG flow equations. First, momenta, Matsubara frequencies, coordinates, and time are rescaled with the scale factor $b = \Lambda_0/\Lambda$ according to their scaling dimensions as

$$\begin{aligned} \mathbf{k}' &= b^{[\mathbf{k}]}\mathbf{k}, & \mathbf{x}' &= b^{-[\mathbf{k}]}\mathbf{x}, \\ \omega'_n &= b^{[\omega_n]}\omega_n, & \tau' &= b^{-[\omega_n]}\tau, \end{aligned} \tag{B.1}$$

with $[\mathbf{k}] = 1$, and $[\omega_n] = z$. The rescaling of the fields in momentum space is given as

$$\Psi'(\omega'_n, \mathbf{k}') = \zeta_b^{-1}\Psi^<(\omega_n, \mathbf{k}), \tag{B.2}$$

with $\zeta_b$ being the field rescaling factor. In contrast to (58), the coordinates are being explicitly rescaled here. As the dimensionless action does not rescale, the scaling of the wave-function renormalization factors is

$$Z'_\tau = b^{-d-2z}\zeta_b^2 Z^<_\tau, \qquad Z'_x = b^{-d-z-2}\zeta_b^2 Z^<_x, \tag{B.3}$$

with $Z^<_x = 1$ as elsewhere in this work. The constraint $Z'_\tau \overset{!}{=} 1$ ensures conservation of the canonical commutation relations for $V = 0$ and keeps the temperature $\beta$ fixed, such that

$$\zeta_b^2 = \frac{b^{d+2z}}{Z^<_\tau}, \qquad Z'_x = \frac{b^{z-2}}{Z^<_\tau}, \tag{B.4}$$

with $l = \ln b$. Thus, the rescaling of the chemical potential and of the wave-function renormalization factor $V$ are

$$\mu' = \frac{b^z\mu^<}{Z^<_\tau}, \qquad V' = \frac{b^{-z}V^<}{Z^<_\tau}. \tag{B.5}$$

The temporal anomalous dimension $\eta_\tau$ is defined as in (64) and relates to the dynamical scaling exponent $z$ via (66). The total and condensate densities are rescaled as

$$n' = b^d n^<, \qquad n'_c = b^d n^<_c, \tag{B.6}$$

with the scaling dimension being equivalent to the engineering dimension $[n]_{\mathrm{eng}} = [n_c]_{\mathrm{eng}} = d$.

The rescaled interaction coupling is

$$g' = \frac{b^\epsilon g^<}{Z^<_\tau}, \tag{B.7}$$

with $\epsilon = z - d$, which is here inferred from the relation $\mu = n_c g$. The scale factors in (B.5) and (B.7) reflect the scaling dimensions of the couplings, which are in general different from the respective engineering dimensions,

$$[\mu] = z + \eta_\tau, \qquad [g] = \epsilon + \eta_\tau, \qquad [V] = -z + \eta_\tau. \tag{B.8}$$

## C  Matsubara summation

In this appendix, we outline how the Matsubara sums encountered in this work can be performed analytically. To this end, we first introduce the notations for the Bose-Einstein distributions

$$n_\mathrm{B}(\omega_k) = \frac{1}{\exp(\beta\omega_k/Z_\tau) - 1}\,, \qquad n_\mathrm{B}^\pm(\omega_k^\pm) = \frac{1}{\exp(\beta\omega_k^\pm) - 1}\,, \tag{C.1}$$

where $\omega_k$ is the dispersion relation for $V = 0$ within the respective phase and $\omega_k^\pm$ is the dispersion in the case of $V \neq 0$ in the symmetry-broken phase (33). In the following, we drop the argument of the Bose-Einstein distributions and employ $n_k \equiv n_\mathrm{B}(\omega_k)$ and $n_k^\pm \equiv n_\mathrm{B}^\pm(\omega_k^\pm)$.

### C.1  Symmetric phase

In the symmetric phase, no wave-function renormalization occurs at one-loop order and thus $Z_\tau^< = 1$ and $V^< = 0$. The propagator thus takes the form

$$G(k) = \frac{1}{\mathrm{i}\omega_n + \omega_k}\,, \tag{C.2}$$

with the free dispersion relation $\omega_k = \varepsilon_k - \mu$, cf. (22).

The sum over a single propagator, which determines the change of $\mu$ (25), results in the Bose-Einstein contribution:

$$\frac{1}{\beta}\sum_{\omega_n} G(k) = n_k\,. \tag{C.3}$$

Here and in the following, summation over the Matsubara frequencies $\omega_n = 2\pi n T$ always goes from $n = -\infty$ to $n = \infty$. The one-loop correction to the four-point coupling $g$ (28) involves Matsubara sums over two propagators, which read

$$\begin{aligned}
\frac{1}{\beta}\sum_{\omega_n} G(k)G(k) &= \beta n_k(1 + n_k)\,, \\
\frac{1}{\beta}\sum_{\omega_n} G(k)G(-k) &= \frac{1 + 2n_k}{2\omega_k}\,.
\end{aligned} \tag{C.4}$$

The main difference between these two sums is that the second one does not vanish in the zero temperature limit. This accounts for the fact that at vanishing temperatures only ladder diagrams, i.e., repeated scatterings between two initial particles, must be taken into account.

### C.2  Condensed phase

In the symmetry-broken phase with $V \neq 0$, besides the normal, also an anomalous propagator must be taken into account:

$$\begin{aligned}
G(k) &= \frac{Z_x\varepsilon_k + \mu - \mathrm{i}Z_\tau\omega_n + V\omega_n^2}{V^2\left[\omega_n^2 + (\omega_k^-)^2\right]\left[\omega_n^2 + (\omega_k^+)^2\right]}\,, \\
\widetilde{G}(k) &= -\frac{\mu}{V^2\left[\omega_n^2 + (\omega_k^-)^2\right]\left[\omega_n^2 + (\omega_k^+)^2\right]}\,.
\end{aligned} \tag{C.5}$$

The dispersion relations are defined in (32) and (33), where we already employed $\mu_1 = \mu_2 \equiv \mu$ as detailed in Sect. 3.2.3.

To shorten the forthcoming equations, let us first introduce a set of auxiliary functions. We start by defining

$$\Omega_k = \frac{1}{V^2\left[\omega_n^2 + (\omega_k^-)^2\right]\left[\omega_n^2 + (\omega_k^+)^2\right]}\,, \tag{C.6}$$

and the general auxiliary function

$$K_{i,j} = \frac{1}{\beta} \sum_{\omega_n} \omega_n^{2j} \Omega_k^i. \tag{C.7}$$

The auxiliary functions $K_{i,j}$ satisfy the relations

$$K_{i,j} = V^2 K_{i+1,j+2} + A_k K_{i+1,j+1} + \omega_k^2 K_{i+1,j},$$
$$\frac{\partial K_{i,j}}{\partial \ln \beta} = -(1+2j) K_{i,j} + 2i \left( A_k K_{i+1,j+1} + 2V^2 K_{i+1,j+2} \right), \tag{C.8}$$

with $A_k = Z_\tau^2 + 2V(Z_x \varepsilon_k + \mu)$. The second relation can straightforwardly be used to express higher-order derivatives of $K_{i,j}$ in terms of the auxiliary functions (C.7). Using the relations (C.8), every auxiliary function with $i > 1$ can be expressed in terms of auxiliary functions and respective derivatives at $i = 1$. For our purposes, only two auxiliary function will be required explicitly:

$$K_{1,0} = \frac{\omega_k^- \left( 1 + 2n_k^+ \right) - \omega_k^+ \left( 1 + 2n_k^- \right)}{2V^2 \omega_k^- \omega_k^+ \left[ (\omega_k^-)^2 - (\omega_k^+)^2 \right]},$$
$$K_{1,1} = \frac{\omega_k^- \left( 1 + 2n_k^- \right) - \omega_k^+ \left( 1 + 2n_k^+ \right)}{2V^2 \left[ (\omega_k^-)^2 - (\omega_k^+)^2 \right]}. \tag{C.9}$$

Their first derivatives read

$$\frac{\partial K_{1,0}}{\partial \ln \beta} = \beta \frac{n_k^- \left( 1 + n_k^- \right) - n_k^+ \left( 1 + n_k^+ \right)}{V^2 \left[ (\omega_k^-)^2 - (\omega_k^+)^2 \right]},$$
$$\frac{\partial K_{1,1}}{\partial \ln \beta} = \beta \frac{(\omega_k^+)^2 n_k^+ \left( 1 + n_k^+ \right) - (\omega_k^-)^2 n_k^- \left( 1 + n_k^- \right)}{V^2 \left[ (\omega_k^-)^2 - (\omega_k^+)^2 \right]}, \tag{C.10}$$

and likewise the second derivatives:

$$\frac{\partial^2 K_{1,0}}{\partial (\ln \beta)^2} = \frac{\partial K_{1,0}}{\partial \ln \beta} + \beta^2 \frac{\omega_k^+ n_k^+ \left( 1 + n_k^+ \right) \left( 1 + 2n_k^+ \right) - (+ \leftrightarrow -)}{V^2 \left[ (\omega_k^-)^2 - (\omega_k^+)^2 \right]},$$
$$\frac{\partial^2 K_{1,1}}{\partial (\ln \beta)^2} = \frac{\partial K_{1,1}}{\partial \ln \beta} + \beta^2 \frac{(\omega_k^-)^3 n_k^- \left( 1 + n_k^- \right) \left( 1 + 2n_k^- \right) - (+ \leftrightarrow -)}{V^2 \left[ (\omega_k^-)^2 - (\omega_k^+)^2 \right]}. \tag{C.11}$$

Here and in the following, for brevity, we indicate by $(+ \leftrightarrow -)$ the respective exchange of signs of the dispersion relations and the Bose-Einstein distributions. In order to derive all auxiliary functions required in this work, we further derive the third derivatives which are

$$\frac{\partial^3 K_{1,0}}{\partial (\ln \beta)^3} = 3 \frac{\partial^2 K_{1,0}}{\partial (\ln \beta)^2} - 2 \frac{\partial K_{1,0}}{\partial \ln \beta} + \beta^3 \frac{(\omega_k^-)^2 n_k^- \left( 1 + n_k^- \right) \left( 1 + 6n_k^- + 6n_k^{-2} \right) - (+ \leftrightarrow -)}{V^2 \left[ (\omega_k^-)^2 - (\omega_k^+)^2 \right]},$$
$$\frac{\partial^3 K_{1,1}}{\partial (\ln \beta)^3} = 3 \frac{\partial^2 K_{1,1}}{\partial (\ln \beta)^2} - 2 \frac{\partial K_{1,1}}{\partial \ln \beta} + \beta^3 \frac{(\omega_k^+)^4 n_k^+ \left( 1 + n_k^+ \right) \left( 1 + 6n_k^+ + 6n_k^{+2} \right) - (+ \leftrightarrow -)}{V^2 \left[ (\omega_k^-)^2 - (\omega_k^+)^2 \right]}. \tag{C.12}$$

Together with the relations (C.8) this allows us to express all the relevant auxiliary functions in terms of Bose-Einstein distributions.

At this point, we can finally evaluate the Matsubara sums, starting with the single-propagator cases:

$$\frac{1}{\beta} \sum_{\omega_n} G(k) = (Z_x \varepsilon_k + \mu) K_{1,0} + V K_{1,1} - \frac{1}{Z_\tau},$$
$$\frac{1}{\beta} \sum_{\omega_n} \widetilde{G}(k) = -\mu K_{1,0}. \tag{C.13}$$

In the case of two propagators, the Matsubara sums with only normal propagators are

$$\frac{1}{\beta}\sum_{\omega_n} G(k)G(k) = K_{1,0} + \mu^2 K_{2,0} - 2Z_\tau^2 K_{2,1}\,,$$

$$\frac{1}{\beta}\sum_{\omega_n} G(k)G(-k) = K_{1,0} + \mu^2 K_{2,0}\,,$$

(C.14)

and the ones with at least one anomalous propagator read

$$\frac{1}{\beta}\sum_{\omega_n} G(k)\widetilde{G}(k) = -\mu\left(Z_x \varepsilon_k + \mu\right)K_{2,0} - \mu V K_{2,1}\,,$$

$$\frac{1}{\beta}\sum_{\omega_n} \widetilde{G}(k)\widetilde{G}(k) = \mu^2 K_{2,0}\,.$$

(C.15)

Together, Eqs. (C.13)–(C.15) yield the relation

$$\frac{\mu}{\beta}\sum_{\omega_n}\left(\widetilde{G}(k)\widetilde{G}(k) - G(k)G(-k)\right) = \frac{1}{\beta}\sum_{\omega_n}\widetilde{G}(k)\,,$$

(C.16)

which proves useful in showing the Hugenholtz-Pines theorem, $\mu_1 = \mu_2$, in (43).

Corrections to the wave-function renormalizations involve first- and second-order frequency-derivatives of the propagators, cf. (55). In terms of the auxiliary functions the latter read

$$\dot{G}(k) = \left(\mathrm{i}Z_\tau + 2V\omega_n\right)\Omega_k - 2\omega_n\left(2V^2\omega_n^2 + A_k\right)\Omega_k G(k)\,,$$

$$\dot{\widetilde{G}}(k) = -2\omega_n\left(2V^2\omega_n^2 + A_k\right)\Omega_k\widetilde{G}(k)\,,$$

(C.17)

and

$$\ddot{G}(k) = 2\Omega_k\left[V - \left(6V^2\omega_n^2 + A_k\right)G(k) - 2\omega_n\left(2V^2\omega_n^2 + A_k\right)\dot{G}(k)\right]\,,$$

$$\ddot{\widetilde{G}}(k) = -2\Omega_k\left[\left(6V^2\omega_n^2 + A_k\right)\widetilde{G}(k) + 2\omega_n\left(2V^2\omega_n^2 + A_k\right)\dot{\widetilde{G}}(k)\right]\,,$$

(C.18)

respectively. These derivatives can now be used when computing the Matsubara sums, starting with the first derivatives of the normal propagator

$$\frac{1}{\beta}\sum_{\omega_n} G(k)\dot{G}(k) = \mathrm{i}Z_\tau\left[\left(Z_x\varepsilon_k + \mu\right)K_{2,0} - 5VK_{2,1}\right]$$

$$- 4\mathrm{i}Z_\tau\left\{\left[Z_\tau^2\left(Z_x\varepsilon_k + \mu\right) + 2V\mu^2\right]K_{3,1} - Z_\tau^2 V K_{3,2}\right\}\,,$$

$$\frac{1}{\beta}\sum_{\omega_n} G(k)\dot{G}(-k) = \mathrm{i}Z_\tau\left[VK_{2,1} - \left(Z_x\varepsilon_k + \mu\right)K_{2,0}\right]\,,$$

(C.19)

and continuing with those of the anomalous propagator

$$\frac{1}{\beta}\sum_{\omega_n} G(k)\dot{\widetilde{G}}(k) = 2\mathrm{i}Z_\tau\mu\left(A_k K_{3,1} + 2V^2 K_{3,2}\right)\,,$$

$$\frac{1}{\beta}\sum_{\omega_n} \widetilde{G}(k)\dot{\widetilde{G}}(k) = 0\,.$$

(C.20)

We furthermore need the Matsubara sums containing second derivatives in order to determine

the flow equation for $V$. For the second derivative of the normal propagator, we find

$$
\frac{1}{\beta}\sum_{\omega_n} G(k)\ddot{G}(k) = -\left(Z_\tau^2 + A_k\right)K_{2,0} + 6V^2 K_{2,1} - 2A_k\mu^2 K_{3,0}
$$
$$
+ 4\left(5Z_\tau^2 A_k + 9V^2\mu^2 - Z_\tau^4\right)K_{3,1} - 24V^2 Z_\tau^2 K_{3,2}
$$
$$
+ 8\left(A_k^2 - 4V^2\omega_k^2\right)\left(\mu^2 K_{4,1} - 2Z_\tau^2 K_{4,2}\right),
$$
$$
\frac{1}{\beta}\sum_{\omega_n} G(k)\ddot{G}(-k) = -\left(Z_\tau^2 + A_k\right)K_{2,0} + 6V^2 K_{2,1}
$$
$$
- 2A_k\mu^2 K_{3,0} + 4\left(A_k^2 - 4V^2\omega_k^2 + 5V^2\mu^2\right)K_{3,1}
$$
$$
+ 8\mu^2\left(A_k^2 - 4V^2\omega_k^2\right)K_{4,1}, \tag{C.21}
$$

and for the anomalous propagator

$$
\frac{1}{\beta}\sum_{\omega_n} G(k)\ddot{\tilde{G}}(k) = 2\mu A_k\left[\left(Z_x\varepsilon_k + \mu\right)K_{3,0} + V K_{3,1}\right] - 20V^2\mu\left[\left(Z_x\varepsilon_k + \mu\right)K_{3,1} + V K_{3,2}\right]
$$
$$
+ 8\mu\left(4V^2\omega_k^2 - A_k^2\right)\left[\left(Z_x\varepsilon_k + \mu\right)K_{4,1} + V K_{4,2}\right], \tag{C.22}
$$
$$
\frac{1}{\beta}\sum_{\omega_n} \tilde{G}(k)\ddot{\tilde{G}}(k) = -2\mu^2\left(A_k K_{3,0} - 10V^2 K_{3,1}\right) + 8\mu^2\left(A_k^2 - 4V^2\omega_k^2\right)K_{4,1}.
$$

# D  Flow of $V$

In Fig. 7 the flow of the wave-function renormalization $V^<(l)$ is shown for three different temperatures in units of the inverse mean-field chemical potential. Within the first distinct interval below $l_1 < 4$, where the feedback of $V$ on the other couplings is neglected, we observe a temperature independent growth. This is succeeded by the second interval $l_1 \leq l < l_2 = 8$ around the healing scale in which the full feedback of $V$ is taken into account and a temperature dependent flow is found. In the asymptotic regime $l \geq l_2$ the wave-function renormalization $V$ continues growing, while having a decreasing $Z_\tau^<$ for $T \neq 0$, cf. Fig. 4. Thus, the action (8) turns approximately SO($d+1$) symmetric for $l \gg l_2$ [43,44]. The observation of a continuously growing $V$ is in contrast to the predicted convergence to $V^*$ in [47] but in agreement with [44]. For $T = 0$ this implies a missing convergence of the microscopic sound velocity $c_s$; however,

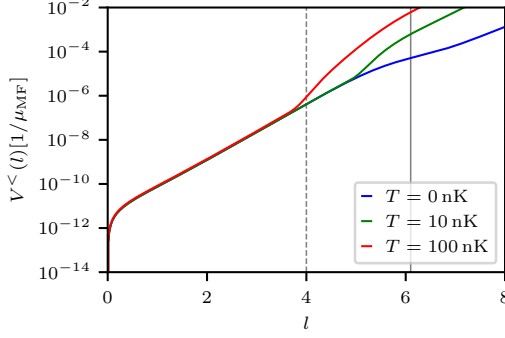

Figure 7: The wave-function renormalization $V^<(l)$ in units of the inverse mean-field chemical potential is plotted against the flow parameter $l$ for three different temperatures. The healing scale is shown by the solid, vertical, gray line, whereas the dashed line separates the first two distinct intervals of the RG flow.

since the coupling $V$ remains much smaller than the mean-field chemical potential its effect on the sound velocity remains negligible. Improving the truncation scheme employed might lead to convergence of the second-order temporal coupling $V$.

Similarly to [44], we do not observe qualitative modifications between $V = 0$ and $V \neq 0$ for the couplings, the condensate depletion, and the shift in critical temperature, cf. Fig. 5 and Fig. 6, respectively. This relates to the fact that in $d = 3$ the IR divergences in our theory only scale logarithmically for $k \to 0$.

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
