# Peer review of "Bogoliubov phonons in a Bose-Einstein condensate from the one-loop perturbative renormalization group"

_SciPost Physics Core, doi:SciPost Phys. Core 7, 066 (2024)_

## Round 1 · Referee Report · Anonymous (Referee 1) · 2024-5-21

Strengths

1- The paper has a clearly defined scope and addresses an interesting research question. 2- The paper is clearly written and very pedagogical. The methods that are being applied are thoroughly explained. Derivations are presented in sufficient detail. Therefore, the paper is accessible also for non-experts. 3- The introduction explains the research context in great detail and provides a useful and extensive guide to the relevant literature.

Report

The paper by Rasch et al. has a well-defined goal, which is stated clearly in the introduction: to present a simple perturbative RG approach to sound excitations in a Bose condensate. This approach is presented in a very clear, pedagogical, and accessible manner. In particular, the results obtained by integrating the RG flow equations demonstrate convincingly that the wave-function renormalization of the temporal derivative terms in the action are the crucial ingredient to describe sound modes and obtain convergent RG flows.

However, comparing the scope of the paper as stated by the authors to the criteria that should be met for a paper to be suitable for publication in SciPost Physics, it seems that the paper does not present the kind of groundbreaking research SciPost Physics is looking for. I believe that the paper is more suitably published in a less selective journal such as SciPost Core.

Requested changes

1- I do not quite understand the sentence below Eq. (7): "To satisfy ..., the linear term must vanish." Is not the vanishing of the linear term the condition that the expansion is around a saddle point? 2- What is the approximation in Eq. (75)? In particular, how does one get from Eq. (69) to Eq. (75)? 3- I believe there is a type below Eq. (75): it should be $1 + \partial_l$.

Recommendation

Accept in alternative Journal (see Report)

  • validity: top
  • significance: good
  • originality: high
  • clarity: top
  • formatting: perfect
  • grammar: perfect

Author:  Niklas Rasch  on 2024-08-05  [id 4679]

(in reply to Report 1 on 2024-05-21)

Dear Referee,

Please find in the attached letter our detailed response to your remarks and questions, as well as a summary of our changes in the manuscript.

With best regards,

Niklas Rasch, Aleksandr N. Mikheev, and Thomas Gasenzer

Attachment:

response_to_editor_v6_XizzxP4.pdf

---

## Round 1 · Referee Report · Anonymous (Referee 2) · 2024-5-22

Strengths

See report

Weaknesses

See report

Report

The authors consider a 3D superfluid within a perturbative one-loop RG approach, focusing on the symmetry-broken phase. They show that the description of the low-energy behavior beyond the healing-length scale requires to include in the action a wavefunction renormalization of the temporal-derivative terms. The condensate depletion is shown to agree with Bogoliubov theory and the interaction-induced shift of the critical temperature is computed.

I do not see any new results in this paper. The problem of infrared divergences appearing in perturbative theory has been thoroughly discussed by Nepomnyashchii and Nepomnyashchii a long time ago and is now well understood in the framework of the functional renormalization group (FRG). In fact, the one-loop approximation proposed by the authors appears to be more complicated than the FRG approach proposed by Wetterich and others, at least when a simple truncation of the effective action is used. The authors justify their approach by the possibility to study universal and non-universal properties of more complex systems such as multi-component Bose gases. There is no doubt that the FRG approach (which has proven very efficient for many strongly-interacting boson systems) would be a much more powerful approach to study such systems.

The goal of the manuscript, which does not clarify any aspects of superfluidity in interacting boson systems, is therefore not clear. A possible goal could have been to show that the FRG flow equations, obtained from a simple truncation of the effective action by Wetterich and others, can be reproduced from a standard one-loop calculation in the symmetry-broken phase (even if I am not fully convinced that this would be enough to justify publication of a paper). Unfortunately, I cannot recommend publication of the manuscript.

More detailed comments are as follows:

1) In the second paragraph of Sec.III, the set of coupling constants $\bf g$ is not defined; it is defined only after Eq.(10).

2) There is a log missing in Eq.(15) and $Z_0[0]$ is missing in the rhs of (16).

3) After Eq.(40), the authors point out that "also the `hole' dispersion $\omega_k^-$ is gapless". Which other dispersion is gapless? Isn't the dispersion $\omega_k^+$ gapped?

4) The way the dimensionless couplings are defined is not always clear. In the FRG approach, dimensionless quantities are defined so as to remove any explicit dependence on the running momentum scale. It seems that a different choice is sometimes made and this obscures a little bit the physical meaning of the dimensionless quantities.

5) The authors ignore the renormalization of the coefficient $Z_x$ of the spatial derivative term, which amounts to neglecting the difference between the condensate density and the superfluid density (since $n_s=Z_x n_c$). This in turn leads to a violation of the Ward identity $n_s=n$ (the full density) associated with Galilean invariance.

6) After Eq.(55), the RG choice $Z_\tau'=1$ is justified by the fact that it preserves the canonical commutation relations of the Bose field. This, however, is not a sufficient condition since the second-order time-derivative term is incompatible with the canonical commutations relations of the Bose field.

7) I do not see how Eq.(63), which relates the dynamical critical exponent $z$ to $\eta_\tau=-\partial_l \ln Z_\tau^<$ can be correct. Since the dynamical critical exponent is defined by $z-1=d/dl c$ where the running sound-mode velocity $c$ depends on both $Z_\tau$ and $V$, the critical exponent $z$ must also depend on $dV/dl$. In fact, it is clear that the result $z=1$ follows from the two following properties: $Z_\tau$ vanishes in the infrared limit while $V$ takes a non-zero value.

8) The asymptotic scaling discussed in Sec.IV.C.1 should be compared with the scaling obtained in the FRG approach; see, e.g., Table I in [45]. More generally, one would like to know whether the one-loop flow equations are fully compatible (or even identical) with the FRG flow equations.

9) At the end of Sec.IV.C.2, the authors explain that they employ different schemes to solve the flow equations: for $l\leq l_1$, they set $V=0$, etc. Why not simply solving the flow equations with no further approximations?

10) The prefactor $\kappa$ of the interaction-induced shift of the critical temperature is an order of magnitude larger than the expected result. The fact that $\kappa$ cannot be reliably obtained from a one-loop approximation should not be a surprise since it requires the knowledge of the one-particle propagator at $T_c$ in a large momentum range including the crossover region between critical and non-critical modes. The calculation has been done using FRG in the BMW approximation with a result $\kappa\simeq 1.37$ which compares well with other methods [F. Benitez {\it et al.}, Phys. Rev. E 85, 026707 (2012)].

11) The coefficient $V_l$ of the second-order temporal derivative term is shown in Fig.7. Why doesn't $V_l$ reach a constant when $l\to\infty$? This behavior doesn't seem compatible with the sound-mode velocity taking a constant value in the infrared limit.

12) The authors never mention the Ginzburg scale $k_G$, whose crucial role has been emphasized by Pistolesi {\it et al.} [41]. Is this non-perturbative scale beyond the reach of the perturbative one-loop approach? Does this explain the apparent poor behavior of $V_l$ mentioned above?

Recommendation

Reject

  • validity: ok
  • significance: low
  • originality: low
  • clarity: good
  • formatting: good
  • grammar: good

Author:  Niklas Rasch  on 2024-08-05  [id 4678]

(in reply to Report 2 on 2024-05-22)

Dear Referee,

Please find in the attached letter our detailed response to your remarks and questions, as well as a summary of our changes in the manuscript.

With best regards,

Niklas Rasch, Aleksandr N. Mikheev, and Thomas Gasenzer

Attachment:

response_to_editor_v6.pdf

---

## Round 2 · Referee Report · Anonymous (Referee 3) · 2024-8-5

Report

The authors have answered all the questions I raised and, in my opinion, also those of the other referee. Therefore, I think that in its present form, the paper is suitable for publication in SciPost Physics Core.

Recommendation

Publish (meets expectations and criteria for this Journal)

---

## Round 2 · Referee Report · Anonymous (Referee 4) · 2024-9-1

Report

I would like to thank the authors for their detailed reply to my first report. They have clarified numerous points and significantly improved their manuscript, and I now recommend publication in SciPost Physics Core.

Recommendation

Publish (meets expectations and criteria for this Journal)

---

## Round 2 · Author Response

Dear Editor,

many thanks for sending us the detailed response of the referees and the editorial recommendation. Herewith, we would like to resubmit our manuscript for publication in SciPost Physics Core. Please find below our detailed response to the referees’ remarks and questions, as well as a summary of our changes in the manuscript.

With best regards,

Niklas Rasch, Aleksandr N. Mikheev, and Thomas Gasenzer

EDITORIAL RECOMMENDATION:

Remarks for authors: The referees do not support publication in SciPost Physics. Based on the reports, publication in SciPost Physics Core may be appropriate after the issues noted in the reports are addressed. For the manuscript to be considered again for SciPost Physics, a resubmitted version would need to make clear how it addresses the points raised by the referees about novelty and significance in the light of previous work.

Requested changes: See the points raised in the two referee reports.

Recommendation: For Journal SciPost Phys. Core: Ask for minor revision

REFEREE 1:

“The authors consider a 3D superfluid within a perturbative one-loop RG approach, focusing on the symmetry-broken phase. They show that the description of the low-energy behavior beyond the healing-length scale requires to include in the action a wavefunction renormalization of the temporal-derivative terms. The condensate depletion is shown to agree with Bogoliubov theory and the interaction-induced shift of the critical temperature is computed.

I do not see any new results in this paper. The problem of infrared divergences appearing in perturbative theory has been thoroughly discussed by Nepomnyashchii and Nepomnyashchii a long time ago and is now well understood in the framework of the functional renormalization group (FRG). In fact, the one-loop approximation proposed by the authors appears to be more complicated than the FRG approach proposed by Wetterich and others, at least when a simple truncation of the effective action is used. The authors justify their approach by the possibility to study universal and non-universal properties of more complex systems such as multi-component Bose gases. There is no doubt that the FRG approach (which has proven very efficient for many strongly-interacting boson systems) would be a much more powerful approach to study such systems.

The goal of the manuscript, which does not clarify any aspects of superfluidity in interacting boson systems, is therefore not clear. A possible goal could have been to show that the FRG flow equations, obtained from a simple truncation of the effective action by Wetterich and others, can be reproduced from a standard one-loop calculation in the symmetry-broken phase (even if I am not fully convinced that this would be enough to justify publication of a paper). Unfortunately, I cannot recommend publication of the manuscript.”

Response: We would like to thank the referee for their careful reading of the manuscript and detailed response concerning our presentation, which have contributed to improving further our manuscript.

Concerning the referee’s general comments: As stated in our manuscript, our scope is to extend upon cited previous discussions on the grounds of the classical Wilson RG and to derive one-loop equations which allow capturing the linear sound-wave away from criticality. We have provided a list of previous literature encompassing different RG methodologies, including the widely studied fRG treatments of different kinds. Our paper is thus meant to close a gap remaining in the basic Wilson-type treatment. A possible goal, to benchmark other approaches, is considered as beyond the scope of the present paper, since that would deviate distinctly from the scope leading to said primary achievements. It thereby remains simple in the sense of being perturbative and evaluated at low loop order, while it is not claimed to be complete, as, e.g., we leave out spatial wave-function renormalization, which is expected to become more relevant close to criticality.

“More detailed comments are as follows:”

Response: Please find below our answers to the points raised by the referee:

1) “In the second paragraph of Sec. III, the set of coupling constants g is not defined; it is defined only after Eq. (10).”

Response: We introduce the set of coupling constants now as Eq. (11), where they are mentioned first, and refer back to this definition below.

2) “There is a log missing in Eq. (15) and Z_0[0] is missing in the rhs of (16).”

Response: We thank the referee for spotting these. The missing logarithm in Eq. (17) as well as Z_0[0] in Eq. (18) are included now.

3) “After Eq. (40), the authors point out that "also the `hole' dispersion ω^−_k is gapless". Which other dispersion is gapless? Isn't the dispersion ω^+_k gapped?”

Response: We deleted this misleading “also” — we only meant to refer to ω_k.

4) “The way the dimensionless couplings are defined is not always clear. In the FRG approach, dimensionless quantities are defined so as to remove any explicit dependence on the running momentum scale. It seems that a different choice is sometimes made and this obscures a little bit the physical meaning of the dimensionless quantities.”

Response: The rescaling procedure employed in our paper is defined in Sect. III C. There, we explicitly describe in detail our rescaling scheme, referring, for comparison, to the standard scheme, provided in App. B, which is commonly employed when considering universal scaling behaviour close to a critical point.

Since, in our work, we aim to extract bulk quantities like the chemical potential at a particular scale which is set, e.g., by the scattering length, we do not rescale the coordinates in the rescaling step of the RG procedure. We emphasise that the choice of the particular rescaling amounts to a rescaling of all couplings involved, which can, in principle, be added after evolving the flow equations. We also remark that a similar scheme has been brought forward in the context of the fRG, cf. Eqs. (A14)f. and (D1) in Ref. [44], where explicit dependencies on the running momentum scale k appear.

5) “The authors ignore the renormalization of the coefficient Z_x of the spatial derivative term, which amounts to neglecting the difference between the condensate density and the superfluid density (since n_s = Z_x n_c). This in turn leads to a violation of the Ward identity n_s = n (the full density) associated with Galilean invariance.”

Response: It is right that we do neglect the wave-function renormalization Z_x of the spatial derivative, for the first and for simplicity of our presentation. In our manuscript, we discuss the flow equations for temperatures T > 0, where the system is not invariant under Galilei transformations and thus the equality between superfluid and total density does not hold due to thermally excited particles. We furthermore include, in our flow equations, the wave-function renormalization coefficient V of the second-order time derivative term in the action. That term ~ \Psi^* V ∂_τ^2\Psi also breaks Galilean symmetry explicitly. Cf. our remarks in the last two paragraphs of Sect. II.B.

We also emphasise that, within the fRG framework, it has been shown that Z_x remains almost unchanged under the RG flow. (Cf. our remark in the end of the last par. of Sect. II.B.)

Hence, the inclusion of Z_x has been left for future work.

We have included further remarks concerning Galilean symmetry of the V-term after Eq. (6) and concerning Z_x at the end of II.B.

6) “After Eq. (55), the RG choice Z′_τ = 1 is justified by the fact that it preserves the canonical commutation relations of the Bose field. This, however, is not a sufficient condition since the second-order time-derivative term is incompatible with the canonical commutations relations of the Bose field.”

Response: As soon as V <> 0, the flow enters the region of quasiparticles and thus modified field operators, such that the commutation relations apply to these. To clarify this point, we added corresponding remarks before what is now Eq. (60).

7) “I do not see how Eq. (63), which relates the dynamical critical exponent z to η_τ = − ∂_l lnZ_τ^< can be correct. Since the dynamical critical exponent is defined by z − 1 = (d/dl)c, where the running sound-mode velocity c depends on both Z_τ and V, the critical exponent z must also depend on dV/dl. In fact, it is clear that the result z = 1 follows from the two following properties: Z_τ vanishes in the infrared limit while V takes a non-zero value.”

Response: It is true that also the second-order time-derivative term ~V ∂_τ^2 in the action should scale as the kinetic term. Nonetheless, defining the dynamical scaling exponent z, in the standard manner, as the scaling dimension of the frequencies [ω] = z, one arrives at (now) Eq. (66).

Evaluating z from the relation between the spatial and temporal second-order derivative terms must, however, remain beyond the scope of our paper, as for non-zero temperatures, V is found to diverge with the flow, and thus any derivation of the dynamical exponent from the computed η_V requires a different approximation level to be chosen in the first place. Hence, we report here on the expected transition to the sound dispersion on the grounds of η_τ, while the corresponding calculation from η_V must remain deferred to future work.

To clarify this point further for the reader, we have added a brief discussion following what is now Eq. (66).

8) “The asymptotic scaling discussed in Sec. IV.C.1 should be compared with the scaling obtained in the FRG approach; see, e.g., Table I in [45]. More generally, one would like to know whether the one-loop flow equations are fully compatible (or even identical) with the FRG flow equations.”

Response: We agree with the referee that a comparison of the resulting flow equations at a certain approximation level would be of further interest. The proposed comparison of asymptotic scalings could be one aspect for comparisons. We consider this task, however, to be beyond the scope of the present manuscript and need to defer it to future work, not at last as the analysis in Ref. [47] has been given for a zero-temperature system.

9) “At the end of Sec. IV.C.2, the authors explain that they employ different schemes to solve the flow equations: for l ≤ l_1, they set V = 0, etc. Why not simply solving the flow equations with no further approximations?”

Response: We thank the referee for pointing out this potential source of confusion. In order to clarify, for the reader, the particular choice of numerical schemes within the different regions of the scale parameter l we have included the second-last paragraph in Sect. IV.C.2. The different schemes allow circumventing the difficulties arising from the numerically large values emerging in the course of the full flow. To counteract these, we have thus chosen approximations which keep numerical inaccuracies and computation time small.

10) “The prefactor κ of the interaction-induced shift of the critical temperature is an order of magnitude larger than the expected result. The fact that κ cannot be reliably obtained from a one-loop approximation should not be a surprise since it requires the knowledge of the one-particle propagator at T_c in a large momentum range including the crossover region between critical and non-critical modes. The calculation has been done using FRG in the BMW approximation with a result κ ≃ 1.37 which compares well with other methods [F. Benitez et al., Phys. Rev. E 85, 026707 (2012)].”

Response: While we were aware of the reasons underlying the observed deviations and had already included a respective discussion in our first submission, we thank the referee for pointing out this work to us. We have included it in our list of references, mentioned it in the introduction section, and added a brief mentioning at the end of Sect. IV.C.3.

11) “The coefficient V_l of the second-order temporal derivative term is shown in Fig. 7. Why doesn't V_l reach a constant when l → ∞? This behavior doesn't seem compatible with the sound-mode velocity taking a constant value in the infrared limit.”

Response: It is right that the coupling for the second-order temporal derivative V does not reach a constant value in the IR. This is in contrast to the result given in Table I in [47] (for the T = 0 case). However, also in Fig. 3 of [44] no convergence of the coupling V is observed. This implies that no convergence of the speed of sound is observed, it must be however pointed out that the coupling V remains small compared to the mean-field chemical potential and thus the effect on the speed of sound remains small. Further, the equality between the microscopic speed of sound and the macroscopic speed of sound only holds for T=0.

To clarify this point, we added a remark in Appendix D.

12) “The authors never mention the Ginzburg scale k_G, whose crucial role has been emphasized by Pistolesi et al. [41]. Is this non-perturbative scale beyond the reach of the perturbative one-loop approach? Does this explain the apparent poor behavior of V_l mentioned above?”

Response: In [35,47] it has been shown that an estimate of the Ginzburg scale can be obtained from a perturbative approximation. However, in [49] it is shown that the scale which governs the crossover from a linear to a quadratic dispersion is the healing-length scale k_xi. Since k_G indicates the scale at which the perturbative Bogoliubov approximation breaks down it also indicates the limitations of a perturbative, one-loop approach as employed in our work.

In order to clarify this point futher, we have included a brief discussion at the end of IV.B.1.

REFEREE 2:

“Strengths: 1- The paper has a clearly defined scope and addresses an interesting research question. 2- The paper is clearly written and very pedagogical. The methods that are being applied are thoroughly explained. Derivations are presented in sufficient detail. Therefore, the paper is accessible also for non-experts. 3- The introduction explains the research context in great detail and provides a useful and extensive guide to the relevant literature.

Report: The paper by Rasch et al. has a well-defined goal, which is stated clearly in the introduction: to present a simple perturbative RG approach to sound excitations in a Bose condensate. This approach is presented in a very clear, pedagogical, and accessible manner. In particular, the results obtained by integrating the RG flow equations demonstrate convincingly that the wave-function renormalization of the temporal derivative terms in the action are the crucial ingredient to describe sound modes and obtain convergent RG flows.

However, comparing the scope of the paper as stated by the authors to the criteria that should be met for a paper to be suitable for publication in SciPost Physics, it seems that the paper does not present the kind of groundbreaking research SciPost Physics is looking for. I believe that the paper is more suitably published in a less selective journal such as SciPost Core.”

Response: We thank the referee for carefully considering our manuscript and for their positive evaluation of our work and presentation. In the following we answer the question raised by the referee.

“Requested changes:” 1- “I do not quite understand the sentence below Eq. (7): "To satisfy ..., the linear term must vanish." Is not the vanishing of the linear term the condition that the expansion is around a saddle point?”

Response: We agree with the referee that these sentences were unclear. We have slightly rewritten the paragraph around (now) Eq. (8).

2- “What is the approximation in Eq. (75)? In particular, how does one get from Eq. (69) to Eq. (75)?”

Response: We thank the referee for pointing to this and clarified the respective step by including a further sentence at the beginning of the paragraph, before what is now Eq. (79).

3- “I believe there is a type below Eq. (75): it should be 1+∂_l.”

Response: We thank the referee for spotting this error! It is corrected now.

---

## Editorial Decision

published